# MIM: Mutual Information Machine

## Abstract

We introduce the Mutual Information Machine (MIM), an autoencoder framework for learning joint distributions over observations and latent states. The model formulation reflects two key design principles: 1) symmetry, to encourage the encoder and decoder to learn different factorizations of the same underlying distribution; and 2) mutual information, to encourage the learning of useful representations for downstream tasks. The objective comprises the Jensen-Shannon divergence between the encoding and decoding joint distributions, plus a mutual information regularizer. We show that this can be bounded by a tractable cross-entropy loss between the true model and a parameterized approximation, and relate this to maximum likelihood estimation and variational autoencoders. Experiments show that MIM is capable of learning a latent representation with high mutual information, and good unsupervised clustering, while providing NLL comparable to VAE (with a sufficiently expressive architecture).

## 1 Introduction

Mutual information is a natural indicator of the quality of a learned representation (Hjelm et al., 2019), along with other characteristics, such as the compositionality of latent factors that are expected to be useful in downstream tasks, like transfer learning (Bengio et al., 2017). Mutual information is, however, computationally difficult to estimate for continuous high-dimensional random variables. As such, it can be hard to optimize when learning latent variable models (Belghazi et al., 2018; Chen et al., 2016a).

This paper formulates a new class of probabilistic autoencoder model that is motivated by two key design principles, namely, the maximization of mutual information, and the symmetry of the encoder-decoder components. Symmetry captures our desire for both the encoder and decoder to effectively and consistently model the underlying observation and latent domains. This is particularly useful for downstream tasks in which either one or both of the encoder and decoder play a central role. These properties are formulated in terms of the symmetric Jensen-Shannon Divergence between the encoder and decoder, combined with an objective term to maximize mutual information. We refer to the resulting model as the *mutual information machine*, or MIM.

We contrast MIM with models trained using (approximate) maximum likelihood, the canonical example being the variational auto-encoder, or VAE (Kingma and Welling, 2013; Rezende et al., 2014). The VAE comprises a probabilistic decoder and an approximate encoder, learned via optimization of an evidence-based lower bound (ELBO) on the log marginal data distribution. In contrast to MIM it is asymmetric in its formulation, and while often producing excellent representations, VAEs sometimes produce pathological results in which the encoder, or approximate posterior, conveys relatively little information between observations and latent states. This behavior, often referred to as *posterior collapse*, results in low mutual information between observations and inferred latent states (Bowman et al., 2015; Chen et al., 2016b; Razavi et al., 2019; van den Oord et al., 2016; 2017).

We formulate the MIM model, and a learning algorithm that minimizes an upper bound on the desired loss. The resulting objective can be viewed as a symmetrized form of KL divergence, thereby closely related to the asymmetric KL objective of the VAE. This also enables direct comparisons to the VAE in terms of posterior collapse, mutual information, data log likelihood, and clustering. Experiments show that MIM offers favourable mutual information, better clustering in the latent representation, and similar reconstruction, at the expense of sampling quality and data log likelihood when compared to VAE with the same architecture. We also demonstrate that for a sufficiently powerful architecture, MIM can match sampling quality and log likelihood of a VAE with the same architecture.

## 2 VARIATIONAL AUTOENCODERS

VAE learning entails optimization of a variational lower bound on the log-marginal likelihood of the data, $\log \mathcal{P}(\boldsymbol{x})$, to estimate the parameters $\theta$ of an approximate posterior $q_{\boldsymbol{\theta}}(\boldsymbol{z}|\boldsymbol{x})$ over latent states $\boldsymbol{z}$ (*i.e.*, the encoder) and a corresponding decoder, $p_{\boldsymbol{\theta}}(\boldsymbol{x}|\boldsymbol{z})$ (Kingma and Welling, 2013; Rezende et al., 2014). A prior over the latent space, $\mathcal{P}(\boldsymbol{z})$, often assumed to be an isotropic Gaussian, serves as a prior for $q_{\boldsymbol{\theta}}(\boldsymbol{z}|\boldsymbol{x})$ in the evidence-lower-bound (ELBO) on the marginal likelihood: $\log \mathcal{P}(\boldsymbol{x}) \geq \mathbb{E}_{\boldsymbol{z} \sim q_{\boldsymbol{\theta}}(\boldsymbol{z}|\boldsymbol{x})} [\log p_{\boldsymbol{\theta}}(\boldsymbol{x}|\boldsymbol{z})] - \mathcal{D}_{\mathrm{KL}} (q_{\boldsymbol{\theta}}(\boldsymbol{z}|\boldsymbol{x}) \| \mathcal{P}(\boldsymbol{z}))$. Here, we use the notation $\mathcal{P}(\boldsymbol{x})$ and $\mathcal{P}(\boldsymbol{z})$ to emphasize that these priors are given, and that we can draw random samples from them, but not necessarily evaluate the log-likelihood of samples under them. In what follows we often refer to them as *anchors* to further emphasize their role.

With amortized posterior inference, we take expectation over the observation distribution, $\mathcal{P}(\boldsymbol{x})$, to obtain the VAE objective:

$$\mathcal{R}_{\mathrm{VAE}}(\boldsymbol{\theta}) = \mathbb{E}_{\boldsymbol{x} \sim \mathcal{P}(\boldsymbol{x})} \left[ \mathbb{E}_{\boldsymbol{z} \sim q_{\boldsymbol{\theta}}(\boldsymbol{z}|\boldsymbol{x})} [\log p_{\boldsymbol{\theta}}(\boldsymbol{x}|\boldsymbol{z})] - \mathcal{D}_{\mathrm{KL}} (q_{\boldsymbol{\theta}}(\boldsymbol{z}|\boldsymbol{x}) \| \mathcal{P}(\boldsymbol{z})) \right]$$
$$= \mathbb{E}_{\boldsymbol{x} \sim \mathcal{P}(\boldsymbol{x}), \boldsymbol{z} \sim q_{\boldsymbol{\theta}}(\boldsymbol{z}|\boldsymbol{x})} [\log p_{\boldsymbol{\theta}}(\boldsymbol{x}|\boldsymbol{z}) + \log \mathcal{P}(\boldsymbol{z}) - \log q_{\boldsymbol{\theta}}(\boldsymbol{z}|\boldsymbol{x})] , \quad (1)$$

Gradients of Eqn. (1) are estimated through MC sampling from $q_{\boldsymbol{\theta}}(\boldsymbol{z}|\boldsymbol{x})$ with reparameterization, yielding unbiased low-variance gradient estimates (Kingma and Welling, 2013; Rezende et al., 2014).

VAEs are normally thought of as maximizing a lower bound on the data log-likelihood, however it can also be expressed as minimizing the divergence between two joint distributions over $\boldsymbol{x}$ and $\boldsymbol{z}$. To see this, we first subtract $\log \mathcal{P}(\boldsymbol{x})$ from (1), which does not change the gradients of the objective with respect to $\theta$. We then negate the result, as we will be performing minimization. This yields

$$\mathcal{L}_{\mathrm{VAE}}(\boldsymbol{\theta}) = -\mathcal{R}_{\mathrm{VAE}}(\boldsymbol{\theta}) + \mathbb{E}_{\boldsymbol{x} \sim \mathcal{P}(\boldsymbol{x})} [\log \mathcal{P}(\boldsymbol{x})] = \mathcal{D}_{\mathrm{KL}} (q_{\boldsymbol{\theta}}(\boldsymbol{z}|\boldsymbol{x})\mathcal{P}(\boldsymbol{x}) \| p_{\boldsymbol{\theta}}(\boldsymbol{x}|\boldsymbol{z})\mathcal{P}(\boldsymbol{z})) . \quad (2)$$

The VAE optimization is therefore equivalent to minimizing the KL divergence between an encoding distribution $q_{\boldsymbol{\theta}}(\boldsymbol{z}|\boldsymbol{x})\mathcal{P}(\boldsymbol{x})$ and a decoding distribution $p_{\boldsymbol{\theta}}(\boldsymbol{x}|\boldsymbol{z})\mathcal{P}(\boldsymbol{z})$.

## 3 SYMMETRY AND MUTUAL INFORMATION

Our goal is to find a consistent encoder-decoder pair, representing a joint distribution over the observation and latent domains, with high mutual information between observations and latent states. By consistent, we mean that the encoding and decoding distributions $q_{\boldsymbol{\theta}}(\boldsymbol{z}|\boldsymbol{x}) \mathcal{P}(\boldsymbol{x})$ and $p_{\boldsymbol{\theta}}(\boldsymbol{x}|\boldsymbol{z}) \mathcal{P}(\boldsymbol{z})$, define the same joint distribution. Effectively, we aim to estimate an undirected graphical model with two valid factorizations. We note that consistency is achievable in the VAE when the approximate posterior $q_{\boldsymbol{\theta}}(\boldsymbol{z}|\boldsymbol{x})$ is capable of representing the posterior under the decoding distribution $p_{\boldsymbol{\theta}}(\boldsymbol{z}|\boldsymbol{x})$. In the general case, however, consistency is not usually achieved.

In contrast to the asymmetric divergence between encoding and decoding distributions in the VAE objective (2), here we consider a symmetric measure, namely, the well-known Jensen-Shannon divergence (JSD),

$$\mathrm{JSD}(\boldsymbol{\theta}) = \frac{1}{2} \Big( \mathcal{D}_{\mathrm{KL}} (p_{\boldsymbol{\theta}}(\boldsymbol{x}|\boldsymbol{z}) \mathcal{P}(\boldsymbol{z}) \| \mathcal{M}_{\mathcal{S}}) + \mathcal{D}_{\mathrm{KL}} (q_{\boldsymbol{\theta}}(\boldsymbol{z}|\boldsymbol{x}) \mathcal{P}(\boldsymbol{x}) \| \mathcal{M}_{\mathcal{S}}) \Big) , \quad (3)$$

where $\mathcal{M}_{\mathcal{S}}$ is an equally weighted mixture of the encoding and decoding distributions; i.e.,

$$\mathcal{M}_{\mathcal{S}} = \frac{1}{2} \big( p_{\boldsymbol{\theta}}(\boldsymbol{x}|\boldsymbol{z}) \mathcal{P}(\boldsymbol{z}) + q_{\boldsymbol{\theta}}(\boldsymbol{z}|\boldsymbol{x}) \mathcal{P}(\boldsymbol{x}) \big) . \quad (4)$$

In addition to encoder-decoder consistency, to learn useful latent representations we also want high mutual information between $\boldsymbol{x}$ and $\boldsymbol{z}$. Indeed, the link between mutual information and representation learning has been explored in recent work (Hjelm et al., 2019; Belghazi et al., 2018; Chen et al., 2016a). Here, to emphasize high mutual information, we add a particular regularizer of the form

$$\mathrm{R}_{\mathrm{H}}(\boldsymbol{\theta}) = \frac{1}{2} \big( H(p_{\boldsymbol{\theta}}(\boldsymbol{x}|\boldsymbol{z}) \mathcal{P}(\boldsymbol{z})) + H(q_{\boldsymbol{\theta}}(\boldsymbol{z}|\boldsymbol{x}) \mathcal{P}(\boldsymbol{x})) \big) . \quad (5)$$

This is the average of the joint entropy over $\boldsymbol{x}$ and $\boldsymbol{z}$ according to the encoding and decoding distributions. This is related to mutual information by the identity $H(\boldsymbol{x}, \boldsymbol{z}) = H(\boldsymbol{x}) + H(\boldsymbol{z}) - I(\boldsymbol{x}; \boldsymbol{z})$. That is, minimizing joint entropy encourages the minimization of the marginal entropy and maximization of the mutual information. In addition to encouraging high mutual information, one can show that this particular regularizer has a deep connection to JSD and the entropy of $\mathcal{M}_{\mathcal{S}}$, i.e.,

$$\mathrm{JSD}(\boldsymbol{\theta}) + \mathrm{R}_{\mathrm{H}}(\boldsymbol{\theta}) = H(\mathcal{M}_{\mathcal{S}}) . \quad (6)$$

The derivation for Eqn. (6) is given in Appendix A.1.

## 4 MUTUAL INFORMATION MACHINE

Section 3 formulates a loss function (6) that reflects our desire for model symmetry and high mutual information. This objective is difficult to optimize directly since we do not know how to evaluate $\log \mathcal{P}(\boldsymbol{x})$ in the general case (*i.e.*, we do not have an exact closed-form expression for $\mathcal{P}(\boldsymbol{x})$). As a consequence, we introduce parameterized approximate priors, $q_{\boldsymbol{\theta}}(\boldsymbol{x})$ and $p_{\boldsymbol{\theta}}(\boldsymbol{z})$, to derive tractable bounds on the penalized Jensen-Shannon divergence. This is similar in spirit to VAEs, which introduce a parameterized approximate posterior. These parameterized priors, together with the conditional encoder and decoder, $q_{\boldsymbol{\theta}}(\boldsymbol{z}|\boldsymbol{x})$ and $p_{\boldsymbol{\theta}}(\boldsymbol{x}|\boldsymbol{z})$, comprise a new pair of joint distributions, $q_{\boldsymbol{\theta}}(\boldsymbol{x}, \boldsymbol{z}) \equiv q_{\boldsymbol{\theta}}(\boldsymbol{z}|\boldsymbol{x}) q_{\boldsymbol{\theta}}(\boldsymbol{x})$, and $p_{\boldsymbol{\theta}}(\boldsymbol{x}, \boldsymbol{z}) \equiv p_{\boldsymbol{\theta}}(\boldsymbol{x}|\boldsymbol{z}) p_{\boldsymbol{\theta}}(\boldsymbol{z})$.

These new joint distributions allow us to formulate a new, tractable loss that bounds $H(\mathcal{M}_{\mathcal{S}})$; i.e.,

$$\mathcal{L}_{\mathrm{CE}}(\boldsymbol{\theta}) \equiv H(\mathcal{M}_{\mathcal{S}}, \mathcal{M}_{\boldsymbol{\theta}}) = \mathcal{D}_{\mathrm{KL}}(\mathcal{M}_{\mathcal{S}} \| \mathcal{M}_{\boldsymbol{\theta}}) + H(\mathcal{M}_{\mathcal{S}}) \geq H(\mathcal{M}_{\mathcal{S}}), \qquad (7)$$

where $\mathcal{M}_{\boldsymbol{\theta}} = \frac{1}{2}(p_{\boldsymbol{\theta}}(\boldsymbol{x}, \boldsymbol{z}) + q_{\boldsymbol{\theta}}(\boldsymbol{x}, \boldsymbol{z}))$, and $H(\mathcal{M}_{\mathcal{S}}, \mathcal{M}_{\boldsymbol{\theta}})$ denotes the cross-entropy between $\mathcal{M}_{\mathcal{S}}$ and $\mathcal{M}_{\boldsymbol{\theta}}$. We refer to $\mathcal{L}_{\mathrm{CE}}$ as the cross-entropy loss. It aims to match the model prior distributions to the anchors, while also minimizing $H(\mathcal{M}_{\mathcal{S}})$. A key advantage of this formulation is that the cross-entropy loss can be trained by Monte Carlo sampling from the anchor distributions with reparameterization (Kingma and Welling, 2013; Rezende et al., 2014).

At this stage it might seem odd to introduce a parametric prior for $\mathcal{P}(\boldsymbol{z})$. Indeed, setting it directly is certainly an option. Nevertheless, in order to achieve consistency between $p_{\boldsymbol{\theta}}(\boldsymbol{x}, \boldsymbol{z})$ and $q_{\boldsymbol{\theta}}(\boldsymbol{x}, \boldsymbol{z})$ it can be advantageous to allow $p_{\boldsymbol{\theta}}(\boldsymbol{z})$ to vary. Essentially, we trade-off latent prior fidelity for increased model consistency. We provide more insights about this in Appendix E.3.

One issue with $\mathcal{L}_{\mathrm{CE}}$ is that, while it will try to enforce consistency between the model and the anchored distributions, i.e., $p_{\boldsymbol{\theta}}(\boldsymbol{x}, \boldsymbol{z}) \approx p_{\boldsymbol{\theta}}(\boldsymbol{x}|\boldsymbol{z})\mathcal{P}(\boldsymbol{z})$ and $q_{\boldsymbol{\theta}}(\boldsymbol{x}, \boldsymbol{z}) \approx q_{\boldsymbol{\theta}}(\boldsymbol{z}|\boldsymbol{x})\mathcal{P}(\boldsymbol{x})$, it will not directly try to achieve model consistency: $p_{\boldsymbol{\theta}}(\boldsymbol{x}, \boldsymbol{z}) \approx q_{\boldsymbol{\theta}}(\boldsymbol{x}, \boldsymbol{z})$. To remedy this, we bound $\mathcal{L}_{\mathrm{CE}}$ using Jensen's inequality, *i.e.*,

$$\mathcal{L}_{\mathrm{MIM}}(\boldsymbol{\theta}) \equiv \frac{1}{2}\big( H(\mathcal{M}_{\mathcal{S}}, q_{\boldsymbol{\theta}}(\boldsymbol{x}, \boldsymbol{z})) + H(\mathcal{M}_{\mathcal{S}}, p_{\boldsymbol{\theta}}(\boldsymbol{x}, \boldsymbol{z})) \big) \geq \mathcal{L}_{\mathrm{CE}}(\boldsymbol{\theta}). \qquad (8)$$

Equation (8) gives us the loss function for the Mutual Information Machine (MIM). It is an average of cross entropy terms between the mixture distribution $\mathcal{M}_{\mathcal{S}}$ and the model encoding and decoding distributions respectively. To see that this encourages model consistency, it can be shown that $\mathcal{L}_{\mathrm{MIM}}$ is equivalent to $\mathcal{L}_{\mathrm{CE}}$ plus a non-negative model consistency regularizer; i.e.,

$$\mathcal{L}_{\mathrm{MIM}}(\boldsymbol{\theta}) = \mathcal{L}_{\mathrm{CE}}(\boldsymbol{\theta}) + \mathrm{R}_{\mathrm{MIM}}(\boldsymbol{\theta}). \qquad (9)$$

The non-negativity of $\mathrm{R}_{\mathrm{MIM}}$ is a simple consequence of $\mathcal{L}_{\mathrm{MIM}}(\boldsymbol{\theta}) \geq \mathcal{L}_{\mathrm{CE}}(\boldsymbol{\theta})$ in (8). One can further show (see Appendix A.2) that $\mathrm{R}_{\mathrm{MIM}}(\boldsymbol{\theta})$ satisfies

$$\mathrm{R}_{\mathrm{MIM}}(\boldsymbol{\theta}) = \frac{1}{2}\big( \mathcal{D}_{\mathrm{KL}}(\mathcal{M}_{\mathcal{S}} \| p_{\boldsymbol{\theta}}(\boldsymbol{x}, \boldsymbol{z})) + \mathcal{D}_{\mathrm{KL}}(\mathcal{M}_{\mathcal{S}} \| q_{\boldsymbol{\theta}}(\boldsymbol{x}, \boldsymbol{z})) \big) - \mathcal{D}_{\mathrm{KL}}(\mathcal{M}_{\mathcal{S}} \| \mathcal{M}_{\boldsymbol{\theta}}). \qquad (10)$$

One can conclude from (10) that the regularizer $\mathrm{R}_{\mathrm{MIM}}$ is zero only when the two joint model distributions, $q_{\boldsymbol{\theta}}(\boldsymbol{x}, \boldsymbol{z})$ and $p_{\boldsymbol{\theta}}(\boldsymbol{x}, \boldsymbol{z})$, are identical under fair samples from the joint sample distribution $\mathcal{M}_{\mathcal{S}}(\boldsymbol{x}, \boldsymbol{z})$. In practice we find that encouraging model consistency also helps stabilize learning.

To understand the MIM objective in greater depth, we find it helpful to express $\mathcal{L}_{\mathrm{MIM}}$ as a sum of fundamental terms that provide some intuition for its expected behavior. In particular, as derived in the Appendix A.3,

$$\mathcal{L}_{\mathrm{MIM}}(\boldsymbol{\theta}) = \mathrm{R}_{\mathrm{H}}(\boldsymbol{\theta}) + \frac{1}{4}\big( \mathcal{D}_{\mathrm{KL}}(\mathcal{P}(\boldsymbol{z}) \| p_{\boldsymbol{\theta}}(\boldsymbol{z})) + \mathcal{D}_{\mathrm{KL}}(\mathcal{P}(\boldsymbol{x}) \| q_{\boldsymbol{\theta}}(\boldsymbol{x})) \big)$$
$$+ \frac{1}{4}\big( \mathcal{D}_{\mathrm{KL}}(q_{\boldsymbol{\theta}}(\boldsymbol{z}|\boldsymbol{x}) \mathcal{P}(\boldsymbol{x}) \| p_{\boldsymbol{\theta}}(\boldsymbol{x}, \boldsymbol{z})) + \mathcal{D}_{\mathrm{KL}}(p_{\boldsymbol{\theta}}(\boldsymbol{x}|\boldsymbol{z}) \mathcal{P}(\boldsymbol{z}) \| q_{\boldsymbol{\theta}}(\boldsymbol{z}, \boldsymbol{x})) \big) \qquad (11)$$

The first term in Eqn. (11) encourages high mutual information between observations and latent states. The second shows that MIM directly encourages the model priors to match the anchor distributions. Indeed, the KL term between the data anchor and the model prior is the maximum likelihood objective. The third term encourages consistency between the model distributions and the anchored distributions, in effect fitting the model decoder to samples drawn from the anchored encoder (cf. VAE), and, via symmetry, fitting the model encoder to samples drawn from the anchored decoder (both with

reparameterization). As such, MIM can be seen as simultaneously training and distilling a model distribution over the data into a latent variable model. The idea of distilling density models has been used in other domains, e.g., for parallelizing auto-regressive models (Oord et al., 2017).

In summary, the MIM loss provides an upper bound on the joint entropy of the observation and latent states under the mixture distribution $\mathcal{M}_\mathcal{S}$: $\mathcal{L}_{\text{MIM}}(\boldsymbol{\theta}) \geq H_{\mathcal{M}_\mathcal{S}}(\boldsymbol{x}) + H_{\mathcal{M}_\mathcal{S}}(\boldsymbol{z}) - I_{\mathcal{M}_\mathcal{S}}(\boldsymbol{x}; \boldsymbol{z})$. Through the MIM loss and the introduction of the parameterized model distribution $\mathcal{M}_\theta$, we are pushing down on the entropy of the anchored mixture distribution $\mathcal{M}_\mathcal{S}$, which is the sum of marginal entropies minus the mutual information. Minimizing the MIM bound yields consistency of the model encoder and decoder, and high mutual information under $\mathcal{M}_\mathcal{S}$ between observations and latent states.

## 5 EXPERIMENTS

In what follows we examine MIM empirically, with the VAE as a baseline. We consider synthetic datasets and well-known image datasets, namely MNIST (LeCun et al., 1998), Fashion-MNIST (Xiao et al., 2017a) and Omniglot (Lake et al., 2015). All models were trained using Adam optimizer Kingma and Ba (2014) with a learning rate of $10^{-3}$, and a mini-batch size of 128. Following Alemi et al. (2017), we anneal the loss to stabilize the optimization. To this end we linearly increase $\beta$ from 0 to 1 in the following expression for a number of 'warm-up' epochs:

$$\hat{\mathcal{L}}_{\text{MIM}}\left(\boldsymbol{\theta}; \{\boldsymbol{x}_i, \boldsymbol{z}_i\}_{i=1}^N, \beta\right) \;=\; -\frac{1}{2N}\sum_{i=1}^N \log\left(p_{\boldsymbol{\theta}}(\boldsymbol{x}_i|\boldsymbol{z}_i)\, q_{\boldsymbol{\theta}}(\boldsymbol{z}_i|\boldsymbol{x}_i)\right) + \beta\left(\log q_{\boldsymbol{\theta}}(\boldsymbol{x}_i)\, p_{\boldsymbol{\theta}}(\boldsymbol{z}_i)\right) . \quad (12)$$

Training continues until the loss (*i.e.*, with $\beta = 1$) on a held-out validation set has not improved for the same number of epochs as the warm-up steps (*i.e.*, defined per experiment). We have found the number of epochs to convergence of MIM learning to be between 2 to 5 times greater than a VAE with the same architecture. A complete description of MIM learning is provided in Appendix C. (Code is available from `https://www.dropbox.com/s/idnls2layat77sj/MIM-master.zip?dl=1` ).

### 5.1 POSTERIOR COLLAPSE IN LOW DIMENSIONAL DATA

Before turning to empirical results, it is useful to briefly revisit similarities and differences between MIM and the canonical VAE formulation. To that end, one can show from Eqn. (1) that the VAE loss can be expressed in a form that bears similarity to the MIM loss in Eqn. (8). In particular, following the derivation in Appendix D,

$$\mathcal{L}_{\text{VAE}} \;=\; \frac{1}{2}\Big( H(\mathcal{M}_\mathcal{S}^{\text{VAE}}, q_{\boldsymbol{\theta}}(\boldsymbol{z}|\boldsymbol{x})\, \mathcal{P}(\boldsymbol{x})) + H(\mathcal{M}_\mathcal{S}^{\text{VAE}}, p_{\boldsymbol{\theta}}(\boldsymbol{x}|\boldsymbol{z})\, \mathcal{P}(\boldsymbol{z})) \Big) - H_{\mathcal{M}_\mathcal{S}^{\text{VAE}}}(\boldsymbol{z}, \boldsymbol{x}) \quad (13)$$

where $\mathcal{M}_\mathcal{S}^{\text{VAE}}(\boldsymbol{x}, \boldsymbol{z}) = q_{\boldsymbol{\theta}}(\boldsymbol{z}|\boldsymbol{x})\, \mathcal{P}(\boldsymbol{x})$, and the entropy in the last term is taken under the sample distribution $\mathcal{M}_\mathcal{S}^{\text{VAE}}$. Like the MIM loss, the first term in Eqn. (13) is the average of two cross-entropy terms, between a sample distribution and the encoding and decoding distributions. Unlike the MIM loss, these terms are asymmetric as samples are drawn only from the encoding distribution. Further, the last term in the VAE loss (13) is equivalent to $-H_{\mathcal{M}_\mathcal{S}^{\text{VAE}}}(\boldsymbol{x}) - H_{\mathcal{M}_\mathcal{S}^{\text{VAE}}}(\boldsymbol{z}) + I_{\mathcal{M}_\mathcal{S}^{\text{VAE}}}(\boldsymbol{x}; \boldsymbol{z})$, and thus it will also encourage a reduction in the mutual information. We posit that this plays a significant role in the phenomena often referred to as posterior collapse, in which the variance of the variational posterior grows large and the latent embedding conveys relatively little information about the observations (e.g., see (Chen et al., 2016b) and others).

To empirically support the expression in Eqn. (13), we consider experiments with synthetic data comprising 2D observations $\boldsymbol{x} \in \mathbb{R}^2$, with a 2D latent space, $\boldsymbol{z} \in \mathbb{R}^2$. In 2D one can easily visualize the model and measure quantitative properties of interest (*e.g.*, mutual information). Fig. 1 depicts data drawn from a Gaussian mixture model comprising five isotropic components with standard deviation 0.25 (top), and latents from an isotropic standard Normal (bottom). The encoder and decoder conditional distributions are Gaussian whose means and variances are regressed from the input using two fully connected layers and *tanh* activation. The parameterized data prior, $q_{\boldsymbol{\theta}}(\boldsymbol{x})$, is defined to be the marginal of the decoding distribution (see Eqn. (23) in Appendix C.1), and the model prior $p_{\boldsymbol{\theta}}(\boldsymbol{z})$ is defined to be the anchor, so the only model parameters are those of the encoder and decoder (see Appendix C for details). We can therefore learn models with the MIM and VAE objectives, but with the same architecture and parameterization. In this section we used a warm-up

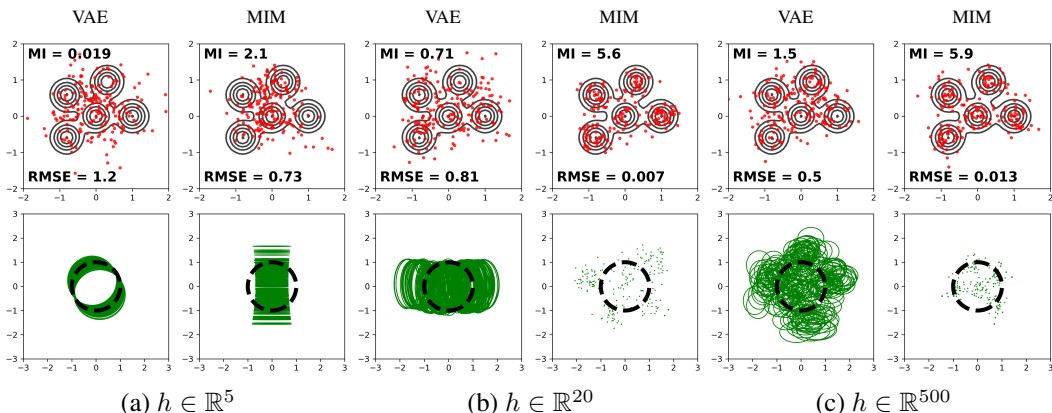

(a) $h \in \mathbb{R}^5$      (b) $h \in \mathbb{R}^{20}$      (c) $h \in \mathbb{R}^{500}$

Figure 1: VAE and MIM models with 2D inputs, a 2D latent space, and 5, 20 and 500 hidden units. Top row: Black contours depict level sets of $\mathcal{P}(\boldsymbol{x})$; red dots are reconstructed test points. Bottom row: Green contours are one standard deviation ellipses of $q_{\boldsymbol{\theta}}(\boldsymbol{z}|\boldsymbol{x})$ for test points. Dashed black circles depict one standard deviation of $\mathcal{P}(\boldsymbol{z})$. The VAE predictive variance remains high, regardless of model expressiveness, an indication of various degrees of posterior collapse, while MIM produces lower predictive variance and lower reconstruction errors, consistent with high mutual information.

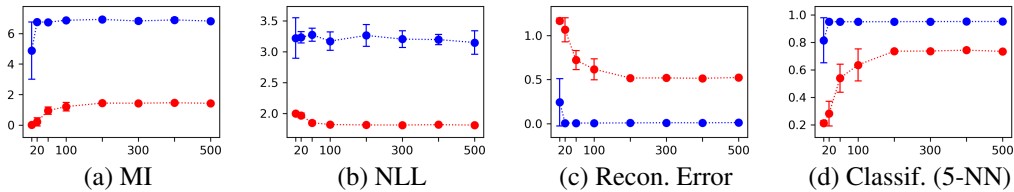

(a) MI      (b) NLL      (c) Recon. Error      (d) Classif. (5-NN)

Figure 2: Test performance for MIM (blue) and VAE (red) for the 2D GMM data (cf. Fig. 1), all as functions of the number of hidden units (on x-axis). Each plot shows the mean and standard deviation of 10 experiments.

scheduler (Vaswani et al., 2017) with a warm-up of 3 steps, and with each epoch comprising 10000 samples. Training and test sets are drawn independently from the GMM.

Figure 1 depicts learned VAE and MIM models, with 5, 20 and 500 hidden units (from left to right) to control model expressiveness. Mutual information and root-mean-squared test reconstruction error are inset in the top row. For the weakest architecture (a), with 5 hidden units, VAE and MIM posterior variances are similar to the prior (MIM in one dimension), a sign of posterior collapse. As the number of hidden units increases (b,c), the VAE posterior variance remains large, preferring lower mutual information while matching the aggregated posterior to the prior. The MIM encoder produces tight posteriors, and yields higher mutual information and lower reconstruction errors.

To quantify this behavior Fig. 2 shows mutual information, the average negative log-likelihood (NLL) of test points under the model $q_{\boldsymbol{\theta}}$, the mean reconstruction error of test points, and 5-NN classification performance[1] (predicting which of 5 GMM components each test point was drawn from), all as functions of the number of hidden units. Following Belghazi et al. (2018), we estimate mutual information using the KSG estimator (Kraskov et al., 2004; Gao et al., 2016), based on 5-NN neighborhoods. The auxiliary classification task provides a proxy for representation quality. Mutual information, reconstruction RMSE, and classification accuracy for test data under the MIM model are better than with VAE models, at the expense of poorer data log likelihoods for MIM. In this case MIM learning finds near-invertible mappings with a sufficiently powerful architecture.

To further investigate MIM learning in low dimensional data, we project 784D images from Fashion-MNIST (Xiao et al., 2017b) onto a 20D linear subspace using PCA (capturing 78.5% of total variance). The training and validation sets had 50,000 and 10,000 images respectively. Fig. 3 summarizes the results, with MIM producing high mutual information and classification accuracy, at all but very low

---

[1]We experimented with 1-NN,3-NN,5-NN,10-NN and found the results to be consistent.

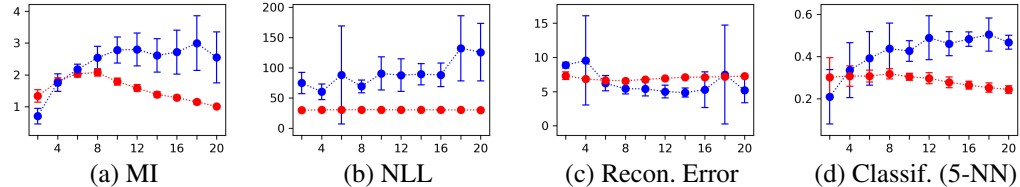

| (a) MI | (b) NLL | (c) Recon. Error | (d) Classif. (5-NN) |

Figure 3: MIM (blue) and VAE (red) for 20D Fashion-MNIST, with latent dimension between 2 and 20. Plots depict mean and standard deviation of 10 experiments.

Table 1: High dimensional image data. Quantitative results based on 10 trials per condition.

|  | convHVAE (S) | | convHVAE (VP) | |
| --- | --- | --- | --- | --- |
| Dataset | MIM | VAE | MIM | VAE |
| Fashion MNIST | $272.14 \pm 0.64$ | $\mathbf{225.40 \pm 0.05}$ | $227.61 \pm 0.34$ | $\mathbf{224.77 \pm 0.04}$ |
| MNIST | $126.85 \pm 0.56$ | $\mathbf{80.50 \pm 0.05}$ | $82.73 \pm 0.08$ | $\mathbf{79.66 \pm 0.06}$ |
| Omniglot | $141.81 \pm 0.32$ | $\mathbf{97.94 \pm 0.29}$ | $104.10 \pm 2.17$ | $\mathbf{97.52 \pm 0.16}$ |

|  | PixelHVAE (S) | | PixelHVAE (VP) | |
| --- | --- | --- | --- | --- |
| Dataset | A-MIM | VAE | A-MIM | VAE |
| Fashion MNIST | $243.95 \pm 0.47$ | $\mathbf{224.65 \pm 0.07}$ | $224.94 \pm 0.34$ | $\mathbf{224.02 \pm 0.08}$ |
| MNIST | $114.96 \pm 0.35$ | $\mathbf{79.04 \pm 0.05}$ | $79.04 \pm 0.08$ | $\mathbf{78.60 \pm 0.04}$ |
| Omniglot | $126.12 \pm 0.38$ | $\mathbf{91.06 \pm 0.14}$ | $91.82 \pm 0.20$ | $\mathbf{90.74 \pm 0.15}$ |

(a) Test NLL (in nats). With a more powerful prior, MIM and VAE yield comparable results.

|  | convHVAE (S) | | convHVAE (VP) | | PixelHVAE (S) | | PixelHVAE (VP) | |
| --- | --- | --- | --- | --- | --- | --- | --- | --- |
| Dataset | MIM | VAE | MIM | VAE | A-MIM | VAE | A-MIM | VAE |
| Fashion MNIST | $\mathbf{0.83}$ | 0.76 | $\mathbf{0.81}$ | 0.78 | 0.71 | $\mathbf{0.76}$ | $\mathbf{0.79}$ | 0.77 |
| MNIST | $\mathbf{0.97}$ | 0.92 | $\mathbf{0.97}$ | 0.92 | $\mathbf{0.95}$ | 0.86 | $\mathbf{0.96}$ | 0.81 |

(b) Test accuracy of 5-NN classifier. Standard deviations are well less than 0.1, and omitted from the table.

latent dimensions. MIM and VAE yield similar test reconstruction errors, with VAE having better negative log likelihoods for test data.

We conclude that the VAE is prone to posterior collapse for a wide range of models' expressiveness and latent dimensionality, with latent embeddings exhibiting low mutual information. In contrast, MIM was empirically robust to posterior collapse. In this regard, we note that several papers have described ways to mitigate posterior collapse in VAE learning, e.g., by lower bounding, or annealing the KL divergence term in the VAE objective (Alemi et al., 2017; Razavi et al., 2019), or by limiting the expressiveness of the decoder (e.g., Chen et al. (2016b)). We posit that MIM does not suffer from this problem as a consequence of the objective design principles that encourage high mutual information between observations and the latent representation.

## 5.2 MIM LEARNING IN HIGH DIMENSIONAL IMAGE DATA

We next consider MIM and VAE learning with image data (Fashion-MNIST, MNIST, Omniglot). Unfortunately, with high dimensional data we cannot reliably compute mutual information (Belghazi et al., 2018). Instead, for model assessment we focus on log-likelihood, reconstruction, and the quality of random samples. In doing so we also explore multiple architectures, including the top performing models from Tomczak and Welling (2017), namely, *convHVAE* (L = 2) and *PixelHVAE* (L = 2), with Standard (S) priors[2], and *VampPrior* (VP) priors[3]. The VP pseudo-inputs are initialized with training data samples. All the experiments below use the same experimental setup as in Tomczak and Welling (2017), and the same latent dimensionality $z \in \mathbb{R}^{80}$. Here we also demonstrate that a powerful prior (*e.g.*, PixelHVAE (VP)) allows MIM to learn models with competitive NLL performance.

---

[2] $p_{\boldsymbol{\theta}}(\boldsymbol{z}) = \mathcal{N}(\boldsymbol{z}; \mu = \mathbf{0}, \sigma = \mathbb{I})$, a standard Normal distribution, where $\mathbb{I}$ is the identity matrix.

[3] $p_{\boldsymbol{\theta}}(\boldsymbol{z}) = \frac{1}{K} \sum_{k=1}^{K} q_{\boldsymbol{\theta}}(\boldsymbol{z}|\boldsymbol{u}_k)$, a mixture model of the encoder conditioned on optimized pseudo-inputs $\boldsymbol{u}_k$.

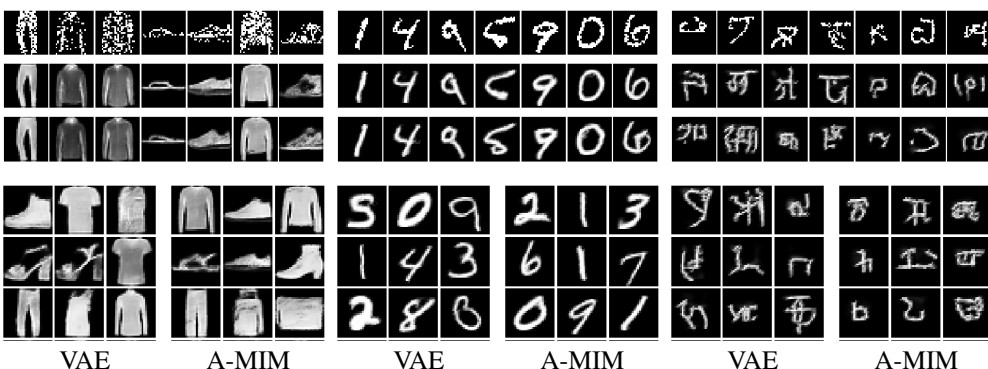

|  VAE | A-MIM | VAE | A-MIM | VAE | A-MIM |

Figure 4: MIM and VAE learning with the PixelHVAE (VP) architecture, applied to Fashion-MNIST, MNIST, and Omniglot (left to right). The top three rows (from top to bottom) are test data samples, VAE reconstruction, and A-MIM reconstruction. Bottom: random samples from VAE and A-MIM.

Sampling from an auto-regressive decoder (*e.g.*, PixelHVAE) is very slow. During training, we find that we can also learn effectively with a sampling distribution comprising just the encoding distribution, i.e., $\mathcal{P}(\boldsymbol{x}) \, q_{\boldsymbol{\theta}}(\boldsymbol{z}|\boldsymbol{x})$, rather than the mixture of the encoder and decoder. We refer to this particular MIM variant as asymmetric MIM (or A-MIM). (For details, see Sec. C.3).

We report quantitative results in Table 1a for test negative log-likelihood (NLL). One can see that VAE models result in better NLL, but with a rather small gap for the more expressive models (*i.e.*, PixelHVAE (VP)). We also show qualitative results for the most expressive models (*i.e.*, PixelHVAE). Fig. 4 depicts reconstruction[4] and sampling for Fashion-MNIST, MNIST, and Omniglot, for the top performing model (PixelHVAE (VP)), with MIM and VAE being comparable. See Appendix F for additional results.

The poor NLL and hence poor sampling for MIM with a weak prior model can be explained by the tightly clustered latent representation (*e.g.*, Fig. 1). A more expressive, learnable prior can capture such clusters more accurately, and as such, also produces good samples (*e.g.*, VampPrior). In other words, while VAE opts for better NLL and sampling at the expense of lower mutual information, MIM provides higher mutual information at the expense of the NLL for a weak prior, and comparable NLL and sampling with more expressive priors.

### 5.3 CLUSTERING AND CLASSIFICATION

Finally, following Hjelm et al. (2019), we consider an auxiliary classification task as a further measure of the quality of the learned representations. We opted for K-NN classification, being a non-parametric method which relies on semantic clustering in latent space, without any additional training. Given representations learned above in Sec. 5.2, a simple 5-NN classifier[5] was applied to test data to predict one of 10 classes for MNIST and Fashion-MNIST.

Table 1b shows that in all but one case, MIM yields more accurate classification results. We attribute the performance difference to higher mutual information of MIM representations. Figure 5 provides a qualitative visualization of the latent clustering, for which t-SNE (van der Maaten and Hinton, 2008)) was used to project the latent space down to 2D. One can see that MIM learning tends to cluster classes in the latent representation more tightly, while VAE clusters are more diffuse and overlapping, consistent with the results in Table 1b.

## 6 RELATED WORK

Given the vast literature on generative models, here we only touch on the major bodies of work related to MIM, beyond the strong connection to VAEs.

As mentioned above, mutual information, together with disentanglement, is considered to be a cornerstone for useful representations (Belghazi et al., 2018; Hjelm et al., 2019). Normalizing flows

---

[4]Test data in the top row of Fig. 4 are binary, while reconstructions depict the probability of each pixel being 1, following Tomczak and Welling (2017)

[5]We omitted results for $k \in \{1, 3, 10\}$ as we find them similar.

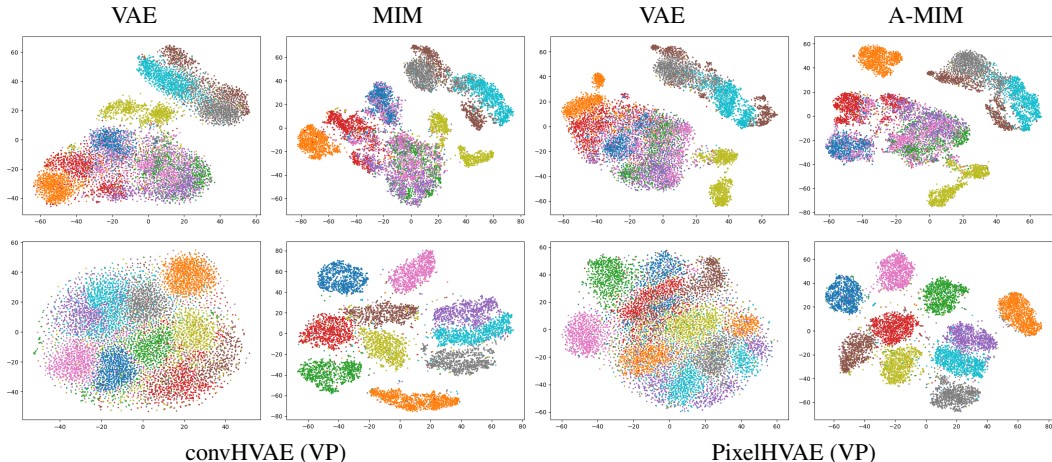

Figure 5: MIM and VAE $z$ embedding for Fashion MNIST (top) and MNIST (botom).

(Rezende and Mohamed, 2015; Dinh et al., 2014; 2016; Kingma and Dhariwal, 2018; Ho et al., 2019) directly maximizes mutual information by restricting the architecture to be invertible and tractable. This, however, requires the latent dimension to be the same as the dimension of the observations (*i.e.*, no bottleneck). As a consequence, normalizing flows are not well suited to learning a concise representation of high dimensional data (*e.g.*, images). Here, MIM often yields mappings that are approximately invertible, with high mutual information and low reconstruction errors.

Bornschein et al. (2015) share some of the same design principles as MIM, i.e., symmetry and encode/decoder consistency. However, their formulation models the joint density in terms of the geometric mean between the encoder and decoder, for which one must compute an expensive partition function. Pu et al. (2017) focuses on minimizing symmetric KL, but must use an adversarial learning procedure, while MIM can be minimized directly.

GANs (Goodfellow et al., 2014), which focus mainly on decoder properties, without a proper inference model, have been shown to minimize JSD between the data anchor $\mathcal{P}(\boldsymbol{x})$ and the model generative process $q_{\boldsymbol{\theta}}(\boldsymbol{x})$ (*i.e.*, the marginal of the decoding distribution in MIM terms). In particular, prior work recognizes the importance of symmetry in learning generative models with reference to symmetric discriminators on $\boldsymbol{x}$ and $\boldsymbol{z}$ (Donahue et al., 2016; Dumoulin et al., 2017; Bang and Shim, 2018). In contrast, here we target JSD between the joint encoding and decoding distributions, together with a regularizer to encourage high mutual information.

## 7  CONCLUSIONS

We introduce a new representation learning framework, named the *mutual information machine* (MIM), that defines a generative model which directly targets high mutual information (*i.e.*, between the observations and the latent representation), and symmetry (*i.e.*, consistency of encoding and decoding factorizations of the joint distribution). We derive a variational bound that enables the maximizion of mutual information in the learned representation for high dimensional continuous data, without the need to directly compute it. We then provide a possible explanation for the phenomena of posterior collapse, and demonstrate that MIM does not suffer from it. Empirical comparisons to VAEs show that MIM learning leads to higher mutual information and better clustering (and classification) in the latent representation, given the same architecture and parametrization. In addition, we show that MIM can provide reconstruction error similar to a deterministic auto-encoder, when the dimensionality of the latent representation is equal to that of the observations. Such behaviour can potentially allow approximate invertibility when the dimensionality differs, with a stochastic mapping that is defined through consistency and high mutual information.

In future work, we intend to focus on utilizing the high mutual information mapping provided by MIM, by exploiting the clustered latent representation to further improve the resulting generative model.

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

# Appendix

## Table of Contents

## A    DERIVATIONS FOR MIM FORMULATION

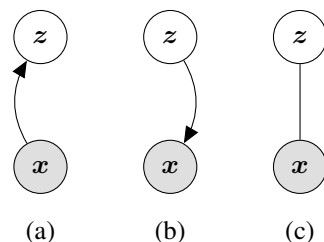

Figure 6: MIM learning estimates two factorizations of a joint distribution: (a) encoding; (b) decoding factorizations. (c) The estimated joint distribution.

In what follows we provide detailed derivations of key elements of the formulation in the paper, namely, Equations (6), (9), and (11). We also consider the relation between MIM based on the Jensen-Shannon divergence and the symmetric KL divergence.

## A.1 Regularized JSD and Entropy

First we develop the relation in Equation (6), between Jensen-Shannon divergence of the encoder and decoder, the average joint entropy of the encoder and decoder, and the joint entropy of the mixture distribution $\mathcal{M}_\mathcal{S}$.

The Jensen-Shannon divergence with respect to the encoding distribution $q_{\boldsymbol{\theta}}(\boldsymbol{z}|\boldsymbol{x})\,\mathcal{P}(\boldsymbol{x})$ and the decoding distribution $p_{\boldsymbol{\theta}}(\boldsymbol{x}|\boldsymbol{z})\,\mathcal{P}(\boldsymbol{z})$ is defined as

$$
\begin{aligned}
\mathrm{JSD}(\boldsymbol{\theta}) &= \frac{1}{2}\left(\mathcal{D}_{\mathrm{KL}}\left(p_{\boldsymbol{\theta}}(\boldsymbol{x}|\boldsymbol{z})\,\mathcal{P}(\boldsymbol{z})\,\|\,\mathcal{M}_\mathcal{S}\right)+\mathcal{D}_{\mathrm{KL}}\left(q_{\boldsymbol{\theta}}(\boldsymbol{z}|\boldsymbol{x})\,\mathcal{P}(\boldsymbol{x})\,\|\,\mathcal{M}_\mathcal{S}\right)\right)\\
&= \frac{1}{2}\Bigg(H(p_{\boldsymbol{\theta}}(\boldsymbol{x}|\boldsymbol{z})\,\mathcal{P}(\boldsymbol{z}),\mathcal{M}_\mathcal{S})-H(p_{\boldsymbol{\theta}}(\boldsymbol{x}|\boldsymbol{z})\,\mathcal{P}(\boldsymbol{z}))\\
&\qquad + H(q_{\boldsymbol{\theta}}(\boldsymbol{z}|\boldsymbol{x})\,\mathcal{P}(\boldsymbol{x}),\mathcal{M}_\mathcal{S})-H(q_{\boldsymbol{\theta}}(\boldsymbol{z}|\boldsymbol{x})\,\mathcal{P}(\boldsymbol{x}))\Bigg)
\end{aligned}
$$

Where $\mathcal{M}_\mathcal{S} = \frac{1}{2}(p_{\boldsymbol{\theta}}(\boldsymbol{x}|\boldsymbol{z})\,\mathcal{P}(\boldsymbol{z})+q_{\boldsymbol{\theta}}(\boldsymbol{z}|\boldsymbol{x})\,\mathcal{P}(\boldsymbol{x}))$ is a mixture of the encoding and decoding distributions. Adding on the regularizer $\mathrm{R}_{\mathrm{H}}(\boldsymbol{\theta}) = \frac{1}{2}(H(p_{\boldsymbol{\theta}}(\boldsymbol{x}|\boldsymbol{z})\,\mathcal{P}(\boldsymbol{z}))+H(q_{\boldsymbol{\theta}}(\boldsymbol{z}|\boldsymbol{x})\,\mathcal{P}(\boldsymbol{x})))$ gives

$$
\begin{aligned}
\mathrm{JSD}(\boldsymbol{\theta})+\mathrm{R}_{\mathrm{H}}(\boldsymbol{\theta}) &= \frac{1}{2}\left(H(p_{\boldsymbol{\theta}}(\boldsymbol{x}|\boldsymbol{z})\,\mathcal{P}(\boldsymbol{z}),\mathcal{M}_\mathcal{S})+H(q_{\boldsymbol{\theta}}(\boldsymbol{z}|\boldsymbol{x})\,\mathcal{P}(\boldsymbol{x}),\mathcal{M}_\mathcal{S})\right)\\
&= H(\mathcal{M}_\mathcal{S})
\end{aligned}
$$

## A.2 MIM Consistency

Here we discuss in greater detail how the learning algorithm encourages consistency between the encoder and decoder of a MIM model, beyond the fact that they are fit to the same sample distribution. To this end we expand on several properties of the model and the optimization procedure.

### A.2.1 MIM consistency regularizer

In what follows we derive the form of the MIM consistency regularizer, $\mathrm{R}_{\mathrm{MIM}}(\boldsymbol{\theta})$, given in Equation (9). Recall that we define $\mathcal{M}_{\boldsymbol{\theta}} = \frac{1}{2}(p_{\boldsymbol{\theta}}(\boldsymbol{x},\boldsymbol{z})+p_{\boldsymbol{\theta}}(\boldsymbol{x},\boldsymbol{z}))$. We can show that $\mathcal{L}_{\mathrm{MIM}}$ is equivalent to $\mathcal{L}_{\mathrm{CE}}$ plus a regularizer by taking their difference.

$$
\begin{aligned}
\mathrm{R}_{\mathrm{MIM}}(\boldsymbol{\theta}) &= \mathcal{L}_{\mathrm{MIM}}(\boldsymbol{\theta})-\mathcal{L}_{\mathrm{CE}}(\boldsymbol{\theta})\\
&= \frac{1}{2}(H(\mathcal{M}_\mathcal{S},p_{\boldsymbol{\theta}}(\boldsymbol{x},\boldsymbol{z}))+H(\mathcal{M}_\mathcal{S},p_{\boldsymbol{\theta}}(\boldsymbol{x},\boldsymbol{z})))-H(\mathcal{M}_\mathcal{S},\mathcal{M}_{\boldsymbol{\theta}})\\
&= \frac{1}{2}(\mathcal{D}_{\mathrm{KL}}\left(\mathcal{M}_\mathcal{S}\,\|\,p_{\boldsymbol{\theta}}(\boldsymbol{x},\boldsymbol{z})\right)+H(\mathcal{M}_\mathcal{S})+\mathcal{D}_{\mathrm{KL}}\left(\mathcal{M}_\mathcal{S}\,\|\,p_{\boldsymbol{\theta}}(\boldsymbol{x},\boldsymbol{z})\right)+H(\mathcal{M}_\mathcal{S}))\\
&\quad -\mathcal{D}_{\mathrm{KL}}\left(\mathcal{M}_\mathcal{S}\,\|\,\mathcal{M}_{\boldsymbol{\theta}}\right)-H(\mathcal{M}_\mathcal{S})\\
&= \frac{1}{2}(\mathcal{D}_{\mathrm{KL}}\left(\mathcal{M}_\mathcal{S}\,\|\,p_{\boldsymbol{\theta}}(\boldsymbol{x},\boldsymbol{z})\right)+\mathcal{D}_{\mathrm{KL}}\left(\mathcal{M}_\mathcal{S}\,\|\,p_{\boldsymbol{\theta}}(\boldsymbol{x},\boldsymbol{z})\right))-\mathcal{D}_{\mathrm{KL}}\left(\mathcal{M}_\mathcal{S}\,\|\,\mathcal{M}_{\boldsymbol{\theta}}\right)
\end{aligned}
$$

where $\mathrm{R}_{\mathrm{MIM}}(\boldsymbol{\theta})$ is non-negative, and is zero only when the encoding and decoding distributions are consistent (*i.e.*, they represent the same joint distribution). To prove that $\mathrm{R}_{\mathrm{MIM}}(\boldsymbol{\theta}) \geq 0$ we now construct Equation (9) in terms of expectation over a joint distribution, which yields

$$
\begin{aligned}
\mathrm{R}_{\mathrm{MIM}}(\boldsymbol{\theta}) &= \frac{1}{2}(H(\mathcal{M}_\mathcal{S},p_{\boldsymbol{\theta}}(\boldsymbol{x},\boldsymbol{z}))+H(\mathcal{M}_\mathcal{S},p_{\boldsymbol{\theta}}(\boldsymbol{x},\boldsymbol{z})))-H(\mathcal{M}_\mathcal{S},\mathcal{M}_{\boldsymbol{\theta}})\\
&= \mathbb{E}_{\boldsymbol{x},\boldsymbol{z}\sim\mathcal{M}_\mathcal{S}}\left[-\frac{1}{2}\log p_{\boldsymbol{\theta}}(\boldsymbol{x},\boldsymbol{z})-\frac{1}{2}\log p_{\boldsymbol{\theta}}(\boldsymbol{x},\boldsymbol{z})+\log\frac{1}{2}(q_{\boldsymbol{\theta}}(\boldsymbol{x},\boldsymbol{z})+p_{\boldsymbol{\theta}}(\boldsymbol{x},\boldsymbol{z}))\right]\\
&= \mathbb{E}_{\boldsymbol{x},\boldsymbol{z}\sim\mathcal{M}_\mathcal{S}}\left[-\log\sqrt{q_{\boldsymbol{\theta}}(\boldsymbol{x},\boldsymbol{z})\cdot p_{\boldsymbol{\theta}}(\boldsymbol{x},\boldsymbol{z})}+\log\frac{1}{2}(q_{\boldsymbol{\theta}}(\boldsymbol{x},\boldsymbol{z})+p_{\boldsymbol{\theta}}(\boldsymbol{x},\boldsymbol{z}))\right]\\
&= \mathbb{E}_{\boldsymbol{x},\boldsymbol{z}\sim\mathcal{M}_\mathcal{S}}\left[-\log\frac{\sqrt{q_{\boldsymbol{\theta}}(\boldsymbol{x},\boldsymbol{z})\cdot p_{\boldsymbol{\theta}}(\boldsymbol{x},\boldsymbol{z})}}{\frac{1}{2}(q_{\boldsymbol{\theta}}(\boldsymbol{x},\boldsymbol{z})+p_{\boldsymbol{\theta}}(\boldsymbol{x},\boldsymbol{z}))}\right] \geq 0
\end{aligned}
$$

where the inequality follows Jensen's inequality, and equality holds only when $q_{\boldsymbol{\theta}}(\boldsymbol{x},\boldsymbol{z}) = p_{\boldsymbol{\theta}}(\boldsymbol{x},\boldsymbol{z})$ (*i.e.*, encoding and decoding distributions are consistent).

### A.2.2 Self-Correcting Gradient

One important property of the optimization follows directly from the difference between the gradient of the upper bound $\mathcal{L}_{\text{MIM}}$ and the gradient of the cross-entropy loss $\mathcal{L}_{\text{CE}}$. By moving the gradient operator into the expectation using reparametrization, one can express the gradient of $\mathcal{L}_{\text{MIM}}(\boldsymbol{\theta})$ in terms of the gradient of $\log \mathcal{M}_{\boldsymbol{\theta}}$ and the regularization term in Equation (9). That is, with some manipulation one obtains

$$\frac{\partial}{\partial \boldsymbol{\theta}} \left( \frac{\log q_{\boldsymbol{\theta}} + \log p_{\boldsymbol{\theta}}}{2} \right) = \frac{\partial}{\partial \boldsymbol{\theta}} \log \left( \frac{q_{\boldsymbol{\theta}} + p_{\boldsymbol{\theta}}}{2} \right) + \frac{1}{2} \frac{\left( \frac{p_{\boldsymbol{\theta}}}{q_{\boldsymbol{\theta}}} - 1 \right) \frac{\partial}{\partial \boldsymbol{\theta}} q_{\boldsymbol{\theta}} + \left( \frac{q_{\boldsymbol{\theta}}}{p_{\boldsymbol{\theta}}} - 1 \right) \frac{\partial}{\partial \boldsymbol{\theta}} p_{\boldsymbol{\theta}}}{q_{\boldsymbol{\theta}} + p_{\boldsymbol{\theta}}} \,, \quad (14)$$

which shows that for any data point where a gap $q_{\boldsymbol{\theta}} > p_{\boldsymbol{\theta}}$ exists, the gradient applied to $p_{\boldsymbol{\theta}}$ grows with the gap, while placing correspondingly less weight on the gradient applied to $q_{\boldsymbol{\theta}}$. The opposite is true when $q_{\boldsymbol{\theta}} < p_{\boldsymbol{\theta}}$. In both case this behaviour encourages consistency between the encoder and decoder. Empirically, we find that the encoder and decoder become reasonably consistent early in the optimization process.

### A.2.3 Numerical Stability

Instead of optimizing an upper bound $\mathcal{L}_{\text{MIM}}$, one might consider a direct optimization of $\mathcal{L}_{\text{CE}}$. Earlier we discussed the importance of the consistency regularizer in $\mathcal{L}_{\text{MIM}}$. Here we motivate the use of $\mathcal{L}_{\text{MIM}}$ from a numerical perspective point of view. In order to optimize $\mathcal{L}_{\text{CE}}$ directly, one must convert $\log q_{\boldsymbol{\theta}}$ and $\log p_{\boldsymbol{\theta}}$ to $q_{\boldsymbol{\theta}}$ and $p_{\boldsymbol{\theta}}$. Unfortunately, this is has the potential to produce numerical errors, especially with 32-bit floating-point precision on GPUs. While various tricks can reduce numerical instability, we find that using the upper bound eliminates the problem while providing the additional benefits outlined above.

### A.2.4 Tractability

As mentioned earlier, there may be several ways to combine the encoder and decoder into a single probabilistic model. One possibility we considered, as an alternative to $\mathcal{M}_{\boldsymbol{\theta}}$ in Equation (7), is

$$\mathcal{M}_{\boldsymbol{\theta}} = \frac{1}{\beta} \sqrt{q_{\boldsymbol{\theta}} \, p_{\boldsymbol{\theta}}} \,, \quad (15)$$

where $\beta = \int \sqrt{q_{\boldsymbol{\theta}} \, p_{\boldsymbol{\theta}}} \, d\boldsymbol{x} \, d\boldsymbol{z}$ is the partition function, similar to Bornschein et al. (2015). One could then define the objective to be the cross-entropy as above with a regularizer to encourage $\beta$ to be close to 1, and hence to encourage consistency between the encoder and decoder. This, however, requires a good approximation to the partition function. Our choice of $\mathcal{M}_{\boldsymbol{\theta}}$ avoids the need for a good value approximation by using reparameterization, which results in unbiased low-variance gradient, independent of the accuracy of the approximation of the value.

### A.3 MIM Loss Decomposition

Here we show how to break down the $\mathcal{L}_{\text{MIM}}$ into the set of intuitive components given in Equation (11). To this end, first note the definition of $\mathcal{L}_{\text{MIM}}$:

$$\mathcal{L}_{\text{MIM}}(\boldsymbol{\theta}) = \frac{1}{2} (H(\mathcal{M}_{\mathcal{S}}, p_{\boldsymbol{\theta}}(\boldsymbol{x}, \boldsymbol{z})) + H(\mathcal{M}_{\mathcal{S}}, p_{\boldsymbol{\theta}}(\boldsymbol{x}, \boldsymbol{z}))) \quad (16)$$

We will focus on the first half of Equation (16) for now,

$$\frac{1}{2} H(\mathcal{M}_{\mathcal{S}}, p_{\boldsymbol{\theta}}(\boldsymbol{x}, \boldsymbol{z})) = \frac{1}{4} \left( H(p_{\boldsymbol{\theta}}(\boldsymbol{x}|\boldsymbol{z}) \, \mathcal{P}(\boldsymbol{z}), p_{\boldsymbol{\theta}}(\boldsymbol{x}, \boldsymbol{z})) + H(q_{\boldsymbol{\theta}}(\boldsymbol{z}|\boldsymbol{x}) \, \mathcal{P}(\boldsymbol{x}), p_{\boldsymbol{\theta}}(\boldsymbol{x}, \boldsymbol{z})) \right) \quad (17)$$

It will be more clear to write out the first term of Equation (17), $\frac{1}{4} H(p_{\boldsymbol{\theta}}(\boldsymbol{x}|\boldsymbol{z}) \, \mathcal{P}(\boldsymbol{z}), p_{\boldsymbol{\theta}}(\boldsymbol{x}, \boldsymbol{z}))$ in full

$$\begin{aligned}
\frac{1}{4} H(p_{\boldsymbol{\theta}}(\boldsymbol{x}|\boldsymbol{z}) \, \mathcal{P}(\boldsymbol{z}), p_{\boldsymbol{\theta}}(\boldsymbol{x}, \boldsymbol{z})) &= -\frac{1}{4} \int_{\boldsymbol{x}, \boldsymbol{z}} p_{\boldsymbol{\theta}}(\boldsymbol{x}|\boldsymbol{z}) \, \mathcal{P}(\boldsymbol{z}) \log(p_{\boldsymbol{\theta}}(\boldsymbol{x}, \boldsymbol{z})) \mathrm{d}\boldsymbol{x} \mathrm{d}\boldsymbol{z} \\
&= -\frac{1}{4} \int_{\boldsymbol{x}, \boldsymbol{z}} p_{\boldsymbol{\theta}}(\boldsymbol{x}|\boldsymbol{z}) \, \mathcal{P}(\boldsymbol{z}) \log(p_{\boldsymbol{\theta}}(\boldsymbol{x}|\boldsymbol{z})) \mathrm{d}\boldsymbol{x} \mathrm{d}\boldsymbol{z} \\
&\quad -\frac{1}{4} \int_{\boldsymbol{x}, \boldsymbol{z}} p_{\boldsymbol{\theta}}(\boldsymbol{x}|\boldsymbol{z}) \, \mathcal{P}(\boldsymbol{z}) \log(p_{\boldsymbol{\theta}}(\boldsymbol{z})) \mathrm{d}\boldsymbol{x} \mathrm{d}\boldsymbol{z}
\end{aligned}$$

We then add and subtract $\frac{1}{4}H(\mathcal{P}(\boldsymbol{z}))$, where

$$\frac{1}{4}H(\mathcal{P}(\boldsymbol{z})) \;=\; -\frac{1}{4}\int_{\boldsymbol{x},\boldsymbol{z}}\mathcal{P}(\boldsymbol{z})\log(\mathcal{P}(\boldsymbol{z}))\mathrm{d}\boldsymbol{x}\mathrm{d}\boldsymbol{z} \;=\; -\frac{1}{4}\int_{\boldsymbol{x},\boldsymbol{z}}p_{\boldsymbol{\theta}}(\boldsymbol{x}|\boldsymbol{z})\,\mathcal{P}(\boldsymbol{z})\log(\mathcal{P}(\boldsymbol{z}))\mathrm{d}\boldsymbol{x}\mathrm{d}\boldsymbol{z}$$

Writing out $\frac{1}{4}H(p_{\boldsymbol{\theta}}(\boldsymbol{x}|\boldsymbol{z})\,\mathcal{P}(\boldsymbol{z}),p_{\boldsymbol{\theta}}(\boldsymbol{x},\boldsymbol{z})) + \frac{1}{4}H(\mathcal{P}(\boldsymbol{z})) - \frac{1}{4}H(\mathcal{P}(\boldsymbol{z}))$ and combining terms, we obtain

$$\frac{1}{4}H(p_{\boldsymbol{\theta}}(\boldsymbol{x}|\boldsymbol{z})\,\mathcal{P}(\boldsymbol{z}),p_{\boldsymbol{\theta}}(\boldsymbol{x},\boldsymbol{z})) \;=\; H(p_{\boldsymbol{\theta}}(\boldsymbol{x}|\boldsymbol{z})\,\mathcal{P}(\boldsymbol{z})) + \mathcal{D}_{\mathrm{KL}}\left(\mathcal{P}(\boldsymbol{z})\,\|\,p_{\boldsymbol{\theta}}(\boldsymbol{z})\right) \qquad (18)$$

The second term in Equation (17) can then be rewritten as

$$\frac{1}{4}H(q_{\boldsymbol{\theta}}(\boldsymbol{z}|\boldsymbol{x})\,\mathcal{P}(\boldsymbol{x}),p_{\boldsymbol{\theta}}(\boldsymbol{x},\boldsymbol{z})) \;=\; \frac{1}{4}\mathcal{D}_{\mathrm{KL}}\left(q_{\boldsymbol{\theta}}(\boldsymbol{z}|\boldsymbol{x})\,\mathcal{P}(\boldsymbol{x})\,\|\,p_{\boldsymbol{\theta}}(\boldsymbol{x},\boldsymbol{z})\right) + \frac{1}{4}H(q_{\boldsymbol{\theta}}(\boldsymbol{z}|\boldsymbol{x})\,\mathcal{P}(\boldsymbol{x})) \quad (19)$$

Combining Equations (18) and (19), we get the interpretable form for Equation 17, i.e.,

$$\begin{aligned}
\frac{1}{2}H(\mathcal{M}_{\mathcal{S}},p_{\boldsymbol{\theta}}(\boldsymbol{x},\boldsymbol{z})) &= \frac{1}{4}\left(H(p_{\boldsymbol{\theta}}(\boldsymbol{x}|\boldsymbol{z})\,\mathcal{P}(\boldsymbol{z})) + H(q_{\boldsymbol{\theta}}(\boldsymbol{z}|\boldsymbol{x})\,\mathcal{P}(\boldsymbol{x}))\right) \\
&\quad + \frac{1}{4}\mathcal{D}_{\mathrm{KL}}\left(\mathcal{P}(\boldsymbol{z})\,\|\,p_{\boldsymbol{\theta}}(\boldsymbol{z})\right) + \frac{1}{4}\mathcal{D}_{\mathrm{KL}}\left(q_{\boldsymbol{\theta}}(\boldsymbol{z}|\boldsymbol{x})\,\mathcal{P}(\boldsymbol{x})\,\|\,p_{\boldsymbol{\theta}}(\boldsymbol{x},\boldsymbol{z})\right) \\
&= \frac{1}{2}\mathrm{R}_{\mathrm{H}}(\boldsymbol{\theta}) + \frac{1}{4}\mathcal{D}_{\mathrm{KL}}\left(\mathcal{P}(\boldsymbol{z})\,\|\,p_{\boldsymbol{\theta}}(\boldsymbol{z})\right) + \frac{1}{4}\mathcal{D}_{\mathrm{KL}}\left(q_{\boldsymbol{\theta}}(\boldsymbol{z}|\boldsymbol{x})\,\mathcal{P}(\boldsymbol{x})\,\|\,p_{\boldsymbol{\theta}}(\boldsymbol{x},\boldsymbol{z})\right)
\end{aligned}$$
$$(20)$$

We can use the same basic steps to derive an analogous expression for $H(\mathcal{M}_{\mathcal{S}},p_{\boldsymbol{\theta}}(\boldsymbol{x},\boldsymbol{z}))$ in Equation (16) and combine it with Equation (20) to get the final interpretable form:

$$\begin{aligned}
\mathcal{L}_{\mathrm{MIM}}(\boldsymbol{\theta}) \;=\;& \mathrm{R}_{\mathrm{H}}(\boldsymbol{\theta}) \;+\; \frac{1}{4}\Big(\mathcal{D}_{\mathrm{KL}}\left(\mathcal{P}(\boldsymbol{z})\,\|\,p_{\boldsymbol{\theta}}(\boldsymbol{z})\right) + \mathcal{D}_{\mathrm{KL}}\left(\mathcal{P}(\boldsymbol{x})\,\|\,q_{\boldsymbol{\theta}}(\boldsymbol{x})\right)\Big) \\
&+ \frac{1}{4}\Big(\mathcal{D}_{\mathrm{KL}}\left(q_{\boldsymbol{\theta}}(\boldsymbol{z}|\boldsymbol{x})\,\mathcal{P}(\boldsymbol{x})\,\|\,p_{\boldsymbol{\theta}}(\boldsymbol{x},\boldsymbol{z})\right) + \mathcal{D}_{\mathrm{KL}}\left(p_{\boldsymbol{\theta}}(\boldsymbol{x}|\boldsymbol{z})\,\mathcal{P}(\boldsymbol{z})\,\|\,q_{\boldsymbol{\theta}}(\boldsymbol{z},\boldsymbol{x})\right)\Big)
\end{aligned}$$

## B  MIM IN TERMS OF SYMMETRIC KL DIVERGENCE

As discussed above, the VAE objective can be expressed as minimizing the KL divergence between the joint anchored encoding and anchored decoding distributions. Below we consider a model formulation using the symmetric KL divergence (SKL),

$$\mathrm{SKL}(\boldsymbol{\theta}) = \frac{1}{2}\left(\mathcal{D}_{\mathrm{KL}}\left(p_{\boldsymbol{\theta}}(\boldsymbol{x}|\boldsymbol{z})\,\mathcal{P}(\boldsymbol{z})\,\|\,q_{\boldsymbol{\theta}}(\boldsymbol{z}|\boldsymbol{x})\,\mathcal{P}(\boldsymbol{x})\right) + \mathcal{D}_{\mathrm{KL}}\left(q_{\boldsymbol{\theta}}(\boldsymbol{z}|\boldsymbol{x})\,\mathcal{P}(\boldsymbol{x})\,\|\,p_{\boldsymbol{\theta}}(\boldsymbol{x}|\boldsymbol{z})\,\mathcal{P}(\boldsymbol{z})\right)\right),$$

the second term of which is the VAE objective. The mutual information regularizer $\mathrm{R}_{\mathrm{H}}(\boldsymbol{\theta})$ given in Equation (5) can be added to SKL to obtain a cross-entropy objective that looks similar to MIM:

$$\frac{1}{2}\mathrm{SKL}(\boldsymbol{\theta}) + \mathrm{R}_{\mathrm{H}}(\boldsymbol{\theta}) = \frac{1}{2}\left(H(\mathcal{M}_{\mathcal{S}},p_{\boldsymbol{\theta}}(\boldsymbol{x}|\boldsymbol{z})\,\mathcal{P}(\boldsymbol{z})) + H(\mathcal{M}_{\mathcal{S}},q_{\boldsymbol{\theta}}(\boldsymbol{z}|\boldsymbol{x})\,\mathcal{P}(\boldsymbol{x}))\right)$$

When the model priors are equal to the anchors, this regularized SKL and MIM are equivalent. In general, however, the MIM loss is not a bound on the regularized SKL.

In what follows we explore the relation between SKL and JSD. In Section A.1 we showed that the Jensen-Shannon divergence can be written as

$$\begin{aligned}
\mathrm{JSD}(\boldsymbol{\theta}) = \frac{1}{2}\Big(&H(p_{\boldsymbol{\theta}}(\boldsymbol{x}|\boldsymbol{z})\,\mathcal{P}(\boldsymbol{z}),\mathcal{M}_{\mathcal{S}}) - H(p_{\boldsymbol{\theta}}(\boldsymbol{x}|\boldsymbol{z})\,\mathcal{P}(\boldsymbol{z})) \\
&+ H(q_{\boldsymbol{\theta}}(\boldsymbol{z}|\boldsymbol{x})\,\mathcal{P}(\boldsymbol{x}),\mathcal{M}_{\mathcal{S}}) - H(q_{\boldsymbol{\theta}}(\boldsymbol{z}|\boldsymbol{x})\,\mathcal{P}(\boldsymbol{x}))\Big) \\
= \frac{1}{2}\Big(&H(p_{\boldsymbol{\theta}}(\boldsymbol{x}|\boldsymbol{z})\,\mathcal{P}(\boldsymbol{z}),\mathcal{M}_{\mathcal{S}}) + H(q_{\boldsymbol{\theta}}(\boldsymbol{z}|\boldsymbol{x})\,\mathcal{P}(\boldsymbol{x}),\mathcal{M}_{\mathcal{S}})\Big) - \mathrm{R}_{\mathrm{H}}(\boldsymbol{\theta})
\end{aligned}$$

Using Jensen's inequality, we can bound $\mathrm{JSD}(\boldsymbol{\theta})$ from above,

$$
\begin{aligned}
\mathrm{JSD}(\boldsymbol{\theta}) &\leq \frac{1}{4}\bigg( H(p_{\boldsymbol{\theta}}(\boldsymbol{x}|\boldsymbol{z})\,\mathcal{P}(\boldsymbol{z})) + H(p_{\boldsymbol{\theta}}(\boldsymbol{x}|\boldsymbol{z})\,\mathcal{P}(\boldsymbol{z}), q_{\boldsymbol{\theta}}(\boldsymbol{z}|\boldsymbol{x})\,\mathcal{P}(\boldsymbol{x})) \\
&\qquad + H(q_{\boldsymbol{\theta}}(\boldsymbol{z}|\boldsymbol{x})\,\mathcal{P}(\boldsymbol{x})) + H(q_{\boldsymbol{\theta}}(\boldsymbol{z}|\boldsymbol{x})\,\mathcal{P}(\boldsymbol{x}), p_{\boldsymbol{\theta}}(\boldsymbol{x}|\boldsymbol{z})\,\mathcal{P}(\boldsymbol{z}))\bigg) - \mathrm{R_H}(\boldsymbol{\theta}) \qquad (21) \\
&= \frac{1}{4}\bigg( H(p_{\boldsymbol{\theta}}(\boldsymbol{x}|\boldsymbol{z})\,\mathcal{P}(\boldsymbol{z}), q_{\boldsymbol{\theta}}(\boldsymbol{z}|\boldsymbol{x})\,\mathcal{P}(\boldsymbol{x})) + H(q_{\boldsymbol{\theta}}(\boldsymbol{z}|\boldsymbol{x})\,\mathcal{P}(\boldsymbol{x}), p_{\boldsymbol{\theta}}(\boldsymbol{x}|\boldsymbol{z})\,\mathcal{P}(\boldsymbol{z})) \\
&\qquad + 2\mathrm{R_H}(\boldsymbol{\theta})\bigg) - \mathrm{R_H}(\boldsymbol{\theta}) \\
&= \frac{1}{4}\bigg( \mathcal{D}_{\mathrm{KL}}\left(p_{\boldsymbol{\theta}}(\boldsymbol{x}|\boldsymbol{z})\,\mathcal{P}(\boldsymbol{z}) \,\|\, q_{\boldsymbol{\theta}}(\boldsymbol{z}|\boldsymbol{x})\,\mathcal{P}(\boldsymbol{x})\right) + \mathcal{D}_{\mathrm{KL}}\left(q_{\boldsymbol{\theta}}(\boldsymbol{z}|\boldsymbol{x})\,\mathcal{P}(\boldsymbol{x}) \,\|\, p_{\boldsymbol{\theta}}(\boldsymbol{x}|\boldsymbol{z})\,\mathcal{P}(\boldsymbol{z})\right) \\
&\qquad + 4\mathrm{R_H}(\boldsymbol{\theta})\bigg) - \mathrm{R_H}(\boldsymbol{\theta}) \\
&= \frac{1}{4}\left( \mathcal{D}_{\mathrm{KL}}\left(p_{\boldsymbol{\theta}}(\boldsymbol{x}|\boldsymbol{z})\,\mathcal{P}(\boldsymbol{z}) \,\|\, q_{\boldsymbol{\theta}}(\boldsymbol{z}|\boldsymbol{x})\,\mathcal{P}(\boldsymbol{x})\right) + \mathcal{D}_{\mathrm{KL}}\left(q_{\boldsymbol{\theta}}(\boldsymbol{z}|\boldsymbol{x})\,\mathcal{P}(\boldsymbol{x}) \,\|\, p_{\boldsymbol{\theta}}(\boldsymbol{x}|\boldsymbol{z})\,\mathcal{P}(\boldsymbol{z})\right)\right) \\
&= \frac{1}{2}\mathrm{SKL}(\boldsymbol{\theta})
\end{aligned}
$$

From Equation (21), if we add the regularizer $\mathrm{R_H}(\boldsymbol{\theta})$ and combine terms, we get

$$
\frac{1}{2}\mathrm{SKL}(\boldsymbol{\theta}) + \mathrm{R_H}(\boldsymbol{\theta}) = \frac{1}{2}\left( H(\mathcal{M}_{\mathcal{S}}, q_{\boldsymbol{\theta}}(\boldsymbol{z}|\boldsymbol{x})\,\mathcal{P}(\boldsymbol{x})) + H(\mathcal{M}_{\mathcal{S}}, p_{\boldsymbol{\theta}}(\boldsymbol{x}|\boldsymbol{z})\,\mathcal{P}(\boldsymbol{z}))\right)
$$

Interestingly, we can write this in terms of KL divergence,

$$
\begin{aligned}
\frac{1}{2}\mathrm{SKL}(\boldsymbol{\theta}) + \mathrm{R_H}(\boldsymbol{\theta}) &= \frac{1}{2}\left( \mathcal{D}_{\mathrm{KL}}\left(\mathcal{M}_{\mathcal{S}} \,\|\, q_{\boldsymbol{\theta}}(\boldsymbol{z}|\boldsymbol{x})\,\mathcal{P}(\boldsymbol{x})\right) + \mathcal{D}_{\mathrm{KL}}\left(\mathcal{M}_{\mathcal{S}} \,\|\, p_{\boldsymbol{\theta}}(\boldsymbol{x}|\boldsymbol{z})\,\mathcal{P}(\boldsymbol{z})\right)\right) + H(\mathcal{M}_{\mathcal{S}}) \\
&= \frac{1}{2}\left( \mathcal{D}_{\mathrm{KL}}\left(\mathcal{M}_{\mathcal{S}} \,\|\, q_{\boldsymbol{\theta}}(\boldsymbol{z}|\boldsymbol{x})\,\mathcal{P}(\boldsymbol{x})\right) + \mathcal{D}_{\mathrm{KL}}\left(\mathcal{M}_{\mathcal{S}} \,\|\, p_{\boldsymbol{\theta}}(\boldsymbol{x}|\boldsymbol{z})\,\mathcal{P}(\boldsymbol{z})\right)\right) \\
&\qquad + \mathrm{JSD}(\boldsymbol{\theta}) + \mathrm{R_H}(\boldsymbol{\theta})
\end{aligned}
$$

which gives the exact relation between JSD and SKL.

$$
\begin{aligned}
\frac{1}{2}\mathrm{SKL}(\boldsymbol{\theta}) &= \frac{1}{2}\left( \mathcal{D}_{\mathrm{KL}}\left(\mathcal{M}_{\mathcal{S}} \,\|\, q_{\boldsymbol{\theta}}(\boldsymbol{z}|\boldsymbol{x})\,\mathcal{P}(\boldsymbol{x})\right) + \mathcal{D}_{\mathrm{KL}}\left(\mathcal{M}_{\mathcal{S}} \,\|\, p_{\boldsymbol{\theta}}(\boldsymbol{x}|\boldsymbol{z})\,\mathcal{P}(\boldsymbol{z})\right)\right) + \mathrm{JSD}(\boldsymbol{\theta}) \\
&= \frac{1}{2}\left( \mathcal{D}_{\mathrm{KL}}\left(\mathcal{M}_{\mathcal{S}} \,\|\, q_{\boldsymbol{\theta}}(\boldsymbol{z}|\boldsymbol{x})\,\mathcal{P}(\boldsymbol{x})\right) + \mathcal{D}_{\mathrm{KL}}\left(\mathcal{M}_{\mathcal{S}} \,\|\, p_{\boldsymbol{\theta}}(\boldsymbol{x}|\boldsymbol{z})\,\mathcal{P}(\boldsymbol{z})\right)\right) \\
&\qquad + \frac{1}{2}\left( \mathcal{D}_{\mathrm{KL}}\left(q_{\boldsymbol{\theta}}(\boldsymbol{z}|\boldsymbol{x})\,\mathcal{P}(\boldsymbol{x}) \,\|\, \mathcal{M}_{\mathcal{S}}\right) + \mathcal{D}_{\mathrm{KL}}\left(p_{\boldsymbol{\theta}}(\boldsymbol{x}|\boldsymbol{z})\,\mathcal{P}(\boldsymbol{z}) \,\|\, \mathcal{M}_{\mathcal{S}}\right)\right)
\end{aligned}
$$

## C  LEARNING

Here we provide a detailed description of MIM learning, with algorithmic pseudo-code. In addition we offer practical considerations regarding the choice of priors' parameterization , and gradient estimation. The empirical upper bound objective is expressed in terms of two cross-entropy terms (cf. (8)). Given $N$ fair samples, $\{\boldsymbol{x}_i, \boldsymbol{z}_i\}_{i=1}^{N}$ drawn from the anchored (sample) distribution, $\mathcal{M}_{\mathcal{S}}(\boldsymbol{x}, \boldsymbol{z})$ in (4), the empirical loss is

$$
\hat{\mathcal{L}}_{\mathrm{MIM}}\left(\boldsymbol{\theta}; \{\boldsymbol{x}_i, \boldsymbol{z}_i\}_{i=1}^{N}\right) \;=\; -\frac{1}{2N}\sum_{i=1}^{N} \log q_{\boldsymbol{\theta}}(\boldsymbol{z}_i|\boldsymbol{x}_i)\, q_{\boldsymbol{\theta}}(\boldsymbol{x}_i) + \log p_{\boldsymbol{\theta}}(\boldsymbol{x}_i|\boldsymbol{z}_i)\, p_{\boldsymbol{\theta}}(\boldsymbol{z}_i)\;, \qquad (22)
$$

where samples from $\mathcal{M}_{\mathcal{S}}$ comprises equal numbers of points from $p_{\boldsymbol{\theta}}(\boldsymbol{x}|\boldsymbol{z})\,\mathcal{P}(\boldsymbol{z})$ and $q_{\boldsymbol{\theta}}(\boldsymbol{z}|\boldsymbol{x})\,\mathcal{P}(\boldsymbol{x})$. Samples from the anchors, $\mathcal{P}(\boldsymbol{x})$ and , $\mathcal{P}(\boldsymbol{z})$, are treated as external observations; i.e., we assume we can sample from them but not necessarily evaluate the density of points under the anchor distributions.

Algorithm 1 specifies the corresponding training procedure. The algorithm makes no assumptions on the form of the parameterized distributions (*e.g.*, discrete, or continuous). In practice, for gradient-based optimization, we would like an unbiased gradient estimator without the need to accurately

approximate the full expectations per se (*i.e.*, in the cross entropy terms). This is particularly important when dealing with high dimensional data (*e.g.*, images), where it is computationally expensive to estimate the value of the expectation. We next discuss practical considerations for the continuous case and the discrete case.

---

**Algorithm 1** MIM learning of parameters $\boldsymbol{\theta}$

---

**Require:** Samples from anchors $\mathcal{P}(\boldsymbol{x}), \mathcal{P}(\boldsymbol{z})$
1: **while** not converged **do**
2:    $D_{\text{dec}} \leftarrow \{\boldsymbol{x}_i, \boldsymbol{z}_i \sim p_{\boldsymbol{\theta}}(\boldsymbol{x}|\boldsymbol{z})\mathcal{P}(\boldsymbol{z})\}_{i=1}^{N/2}$
3:    $D_{\text{enc}} \leftarrow \{\boldsymbol{x}_j, \boldsymbol{z}_j \sim q_{\boldsymbol{\theta}}(\boldsymbol{z}|\boldsymbol{x})\mathcal{P}(\boldsymbol{x})\}_{j=1}^{N/2}$
4:    $D \leftarrow D_{\text{dec}} \bigcup D_{\text{enc}}$
5:    *# See definition of $\hat{\mathcal{L}}_{MIM}$ in Eq. (22)*
6:    $\mathcal{L}_{\text{MIM}}(\boldsymbol{\theta}) \approx \hat{\mathcal{L}}_{\text{MIM}}(\boldsymbol{\theta}; D)$
7:    *# Minimize loss*
8:    $\Delta\boldsymbol{\theta} \propto -\nabla_{\boldsymbol{\theta}} \hat{\mathcal{L}}_{\text{MIM}}(\boldsymbol{\theta}; D)$
9: **end while**

---

## C.1    MIM PARAMETRIC PRIORS

There are several effective ways to parameterize the priors. For the 1D experiments below we model $p_{\boldsymbol{\theta}}(\boldsymbol{z})$ using linear mixtures of isotropic Gaussians. With complex, high dimensional data one might also consider more powerful models (*e.g.*, autoregressive, or flow-based priors). Unfortunately, the use of complex models typically increases the required computational resources, and the training and inference time. As an alternative we use for image data the *vampprior* Tomczak and Welling (2017), which models the latent prior as a mixture of posteriors, i.e., $p_{\boldsymbol{\theta}}(\boldsymbol{z}) = \sum_{k=1}^{K} q_{\boldsymbol{\theta}}(\boldsymbol{z}|\boldsymbol{x} = \boldsymbol{u}_k)$ with learnable pseudo-inputs $\{\boldsymbol{u}_k\}_{k=1}^{K}$. This is effective and allows one to avoid the need for additional parameters (see Tomczak and Welling (2017) for details on vampprior's effect over gradient estimation).

Another useful model with high dimensional data, following Bornschein et al. (2015), is to define $q_{\boldsymbol{\theta}}(\boldsymbol{x})$ as the marginal of the decoding distribution; i.e.,

$$q_{\boldsymbol{\theta}}(\boldsymbol{x}) = \mathbb{E}_{\boldsymbol{z} \sim p_{\boldsymbol{\theta}}(\boldsymbol{z})}\left[p_{\boldsymbol{\theta}}(\boldsymbol{x}|\boldsymbol{z})\right] . \tag{23}$$

Like the vampprior, this entails no new parameters. It also helps to encourage consistency between the encoding and decoding distributions. In addition it enables direct empirical comparisons of VAE learning to MIM learning, because we can then use identical parameterizations and architectures for both. During learning, when $q_{\boldsymbol{\theta}}(\boldsymbol{x})$ is defined as the marginal (23), we evaluate $\log q_{\boldsymbol{\theta}}(\boldsymbol{x})$ with a single sample and reparameterization. When $\boldsymbol{z}$ is drawn directly from the latent prior:

$$\log q_{\boldsymbol{\theta}}(\boldsymbol{x}) = \log \mathbb{E}_{\tilde{\boldsymbol{z}} \sim p_{\boldsymbol{\theta}}(\boldsymbol{z})}\left[p_{\boldsymbol{\theta}}(\boldsymbol{x}|\tilde{\boldsymbol{z}})\right] \approx \log p_{\boldsymbol{\theta}}(\boldsymbol{x}|\tilde{\boldsymbol{z}}) .$$

When $\boldsymbol{z}$ is drawn from the encoder, given a sample observation, we use importance sampling:

$$\log q_{\boldsymbol{\theta}}(\boldsymbol{x}) = \log \mathbb{E}_{\tilde{\boldsymbol{z}} \sim q_{\boldsymbol{\theta}}(\boldsymbol{z}|\boldsymbol{x})}\left[p_{\boldsymbol{\theta}}(\boldsymbol{x}|\tilde{\boldsymbol{z}})\frac{p_{\boldsymbol{\theta}}(\tilde{\boldsymbol{z}})}{q_{\boldsymbol{\theta}}(\tilde{\boldsymbol{z}}|\boldsymbol{x})}\right] \approx \log p_{\boldsymbol{\theta}}(\boldsymbol{x}|\tilde{\boldsymbol{z}}) + \log p_{\boldsymbol{\theta}}(\tilde{\boldsymbol{z}}) - \log q_{\boldsymbol{\theta}}(\tilde{\boldsymbol{z}}|\boldsymbol{x})$$

Algorithm 2 provides algorithm details with the marginal prior.

## C.2    GRADIENT ESTIMATION

Optimization is performed through minibatch stochastic gradient descent. To ensure unbiased gradient estimates of $\hat{\mathcal{L}}_{\text{MIM}}$ we use the reparameterization trick Kingma and Welling (2013); Rezende et al. (2014) when taking expectation with respect to continuous encoder and decoder distributions, $q_{\boldsymbol{\theta}}(\boldsymbol{z}|\boldsymbol{x})$ and $p_{\boldsymbol{\theta}}(\boldsymbol{x}|\boldsymbol{z})$. Reparameterization entails sampling an auxiliary variable $\epsilon \sim p(\epsilon)$, with known $p(\epsilon)$, followed by a deterministic mapping from sample variates to the target random variable, that is $p_{\boldsymbol{\theta}}(\boldsymbol{z}) = g_{\boldsymbol{\theta}}(\epsilon)$ and $q_{\boldsymbol{\theta}}(\boldsymbol{z}|\boldsymbol{x}) = h_{\boldsymbol{\theta}}(\epsilon, \boldsymbol{x})$ for prior and conditional distributions. In doing so we assume $p(\epsilon)$ is independent of the parameters $\boldsymbol{\theta}$. It then follows that

$$\nabla_{\boldsymbol{\theta}} \mathbb{E}_{\boldsymbol{z} \sim q_{\boldsymbol{\theta}}(\boldsymbol{z}|\boldsymbol{x})}\left[f_{\boldsymbol{\theta}}(\boldsymbol{z})\right] = \nabla_{\boldsymbol{\theta}} \mathbb{E}_{\epsilon \sim p(\epsilon)}\left[f_{\boldsymbol{\theta}}(h_{\boldsymbol{\theta}}(\epsilon, \boldsymbol{x}))\right] = \mathbb{E}_{\epsilon \sim p(\epsilon)}\left[\nabla_{\boldsymbol{\theta}} f_{\boldsymbol{\theta}}(h_{\boldsymbol{\theta}}(\epsilon, \boldsymbol{x}))\right]$$

---

**Algorithm 2** MIM learning with marginal $q_{\boldsymbol{\theta}}(\boldsymbol{x})$

---

**Require:** Samples from anchors $\mathcal{P}(\boldsymbol{x}), \mathcal{P}(\boldsymbol{z})$
**Require:** Define $q_{\boldsymbol{\theta}}(\boldsymbol{x}) = \mathbb{E}_{\boldsymbol{z} \sim p_{\boldsymbol{\theta}}(\boldsymbol{z})} [p_{\boldsymbol{\theta}}(\boldsymbol{x}|\boldsymbol{z})]$
1: **while** not converged **do**
2:     *# Sample encoding distribution*
3:     $D_{\mathrm{enc}} \leftarrow \{\boldsymbol{x}_i, \boldsymbol{z}_i \sim q_{\boldsymbol{\theta}}(\boldsymbol{z}|\boldsymbol{x})\mathcal{P}(\boldsymbol{x})\}_{i=1}^N$
4:     *# Compute objective, approximate $\log q_{\boldsymbol{\theta}}(\boldsymbol{x})$ with 1 sample and importance sampling*
5:     $\log q_{\boldsymbol{\theta}}(\boldsymbol{x}_i) \approx \log p_{\boldsymbol{\theta}}(\boldsymbol{x}_i|\boldsymbol{z}_i) + \log p_{\boldsymbol{\theta}}(\boldsymbol{z}_i) - \log q_{\boldsymbol{\theta}}(\boldsymbol{z}_i|\boldsymbol{x}_i)$
6:     $\hat{\mathcal{L}}_{\mathrm{MIM}}(\boldsymbol{\theta}; D_{\mathrm{enc}}) \leftarrow -\frac{1}{N} \sum_{i=1}^N (\log p_{\boldsymbol{\theta}}(\boldsymbol{x}_i|\boldsymbol{z}_i) + \log p_{\boldsymbol{\theta}}(\boldsymbol{z}_i))$
7:     *# Sample decoding distribution*
8:     $D_{\mathrm{dec}} \leftarrow \{\boldsymbol{x}_i, \boldsymbol{z}_i \sim p_{\boldsymbol{\theta}}(\boldsymbol{x}|\boldsymbol{z})\mathcal{P}(\boldsymbol{z})\}_{i=1}^N$
9:     *# Compute objective, approximate $\log q_{\boldsymbol{\theta}}(\boldsymbol{x})$ with 1 sample*
10:    $\log q_{\boldsymbol{\theta}}(\boldsymbol{x}_i) \approx \log p_{\boldsymbol{\theta}}(\boldsymbol{x}_i|\boldsymbol{z}_i)$
11:    $\hat{\mathcal{L}}_{\mathrm{MIM}}(\boldsymbol{\theta}; D_{\mathrm{dec}}) \leftarrow -\frac{1}{2N} \sum_{i=1}^N (\log q_{\boldsymbol{\theta}}(\boldsymbol{z}_i|\boldsymbol{x}_i) + 2 \log p_{\boldsymbol{\theta}}(\boldsymbol{x}_i|\boldsymbol{z}_i) + \log p_{\boldsymbol{\theta}}(\boldsymbol{z}_i))$
12:    *# Minimize loss*
13:    $\Delta\boldsymbol{\theta} \propto -\nabla_{\boldsymbol{\theta}} \frac{1}{2} \left( \hat{\mathcal{L}}_{\mathrm{MIM}}(\boldsymbol{\theta}; D_{\mathrm{dec}}) + \hat{\mathcal{L}}_{\mathrm{MIM}}(\boldsymbol{\theta}; D_{\mathrm{enc}}) \right)$
14: **end while**

---

where $f_{\boldsymbol{\theta}}(\boldsymbol{z})$ is the loss function with parameters $\boldsymbol{\theta}$. It is common to let $p(\epsilon)$ be standard normal, $\epsilon \sim \mathcal{N}(0,1)$, and for $\boldsymbol{z}|\boldsymbol{x}$ to be Gaussian with mean $\mu_{\boldsymbol{\theta}}(\boldsymbol{x})$ and standard deviation $\sigma_{\boldsymbol{\theta}}(\boldsymbol{x})$, in which case $\boldsymbol{z} = \sigma_{\boldsymbol{\theta}}(\boldsymbol{x})\,\epsilon + \mu_{\boldsymbol{\theta}}(\boldsymbol{x})$, A more generic exact density model can be learned by mapping a known base distribution (*e.g.*, Gaussian) to a target distribution with normalizing flows Dinh et al. (2014; 2016); Rezende and Mohamed (2015).

In the case of discrete distributions, e.g., with discrete data, reparameterization is not readily applicable. There exist continuous relaxations that permit reparameterization (e.g., Maddison et al. (2016); Tucker et al. (2017)), but current methods are rather involved in practice, and require adaptation of the objective function or the optimization process. Here we simply use the REINFORCE algorithm Sutton et al. (1999) for unbiased gradient estimates, as follows

$$\nabla_{\boldsymbol{\theta}} \mathbb{E}_{\boldsymbol{z} \sim q_{\boldsymbol{\theta}}(\boldsymbol{z}|\boldsymbol{x})} [f_{\boldsymbol{\theta}}(\boldsymbol{z})] = \mathbb{E}_{\boldsymbol{z} \sim q_{\boldsymbol{\theta}}(\boldsymbol{z}|\boldsymbol{x})} [\nabla_{\boldsymbol{\theta}} f_{\boldsymbol{\theta}}(\boldsymbol{z}) + f_{\boldsymbol{\theta}}(\boldsymbol{z})\nabla_{\boldsymbol{\theta}} \log q_{\boldsymbol{\theta}}(\boldsymbol{z}|\boldsymbol{x})] .$$

A detailed derivation follows the use of the relation below,

$$\nabla_{\boldsymbol{\theta}} q_{\boldsymbol{\theta}}(\boldsymbol{z}|\boldsymbol{x}) = q_{\boldsymbol{\theta}}(\boldsymbol{z}|\boldsymbol{x})\nabla_{\boldsymbol{\theta}} \log q_{\boldsymbol{\theta}}(\boldsymbol{z}|\boldsymbol{x})$$

in order to provide unbiased gradient estimates as follows

$$\begin{aligned}
\nabla_{\boldsymbol{\theta}} \mathbb{E}_{\boldsymbol{z} \sim q_{\boldsymbol{\theta}}(\boldsymbol{z}|\boldsymbol{x})} [f_{\boldsymbol{\theta}}(\boldsymbol{z})] &= \nabla_{\boldsymbol{\theta}} \int f_{\boldsymbol{\theta}}(\boldsymbol{z})\, q_{\boldsymbol{\theta}}(\boldsymbol{z}|\boldsymbol{x})\, d\boldsymbol{z} \\
&= \int q_{\boldsymbol{\theta}}(\boldsymbol{z}|\boldsymbol{x})\, \nabla_{\boldsymbol{\theta}} f_{\boldsymbol{\theta}}(\boldsymbol{z})\, d\boldsymbol{z} + \int f_{\boldsymbol{\theta}}(\boldsymbol{z})\, \nabla_{\boldsymbol{\theta}} q_{\boldsymbol{\theta}}(\boldsymbol{z}|\boldsymbol{x})\, d\boldsymbol{z} \\
&= \int q_{\boldsymbol{\theta}}(\boldsymbol{z}|\boldsymbol{x})\, \nabla_{\boldsymbol{\theta}} f_{\boldsymbol{\theta}}(\boldsymbol{z})\, d\boldsymbol{z} + \int f_{\boldsymbol{\theta}}(\boldsymbol{z})\, q_{\boldsymbol{\theta}}(\boldsymbol{z}|\boldsymbol{x})\, \nabla_{\boldsymbol{\theta}} \log q_{\boldsymbol{\theta}}(\boldsymbol{z}|\boldsymbol{x})\, d\boldsymbol{z} \\
&= \mathbb{E}_{\boldsymbol{z} \sim q_{\boldsymbol{\theta}}(\boldsymbol{z}|\boldsymbol{x})} [\nabla_{\boldsymbol{\theta}} f_{\boldsymbol{\theta}}(\boldsymbol{z}) + f_{\boldsymbol{\theta}}(\boldsymbol{z})\nabla_{\boldsymbol{\theta}} \log q_{\boldsymbol{\theta}}(\boldsymbol{z}|\boldsymbol{x})]
\end{aligned}$$

which facilitate the use of samples to approximate the integral.

## C.3    TRAINING TIME

Training times of MIM models are comparable to training times for VAEs with comparable architectures. One important difference concerns the time required for sampling from the decoder during training. This is particularly significant for models like auto-regressive decoders (*e.g.*, Kingma et al. (2016)) for which sampling is very slow. In such cases, we find that we can also learn effectively with a sampling distribution that only includes samples from the encoding distribution, i.e., $\mathcal{P}(\boldsymbol{x})\, q_{\boldsymbol{\theta}}(\boldsymbol{z}|\boldsymbol{x})$, rather than the mixture. We refer to this particular MIM variant as asymmetric MIM (or A-MIM). We use it in Sec. 5.2 when working with the PixelHVAE architecture Kingma et al. (2016).

## D  POSTERIOR COLLAPSE IN VAE

Here we discuss a possible root cause for the observed phenomena of posterior collapse, and show that VAE learning can be viewed as an asymmetric MIM learning with a regularizer that encourages the appearance of the collapse. We further support that idea in the experiments in Section 5.1. As discussed earlier, VAE learning entails maximization of a variational lower bound (ELBO) on the log-marginal likelihood, or equivalently, given Equation (1), the VAE loss in terms of expectation over a joint distribution:

$$-\mathbb{E}_{\boldsymbol{x}\sim\mathcal{P}(\boldsymbol{x}),\boldsymbol{z}\sim q_{\boldsymbol{\theta}}(\boldsymbol{z}|\boldsymbol{x})}\left[\log p_{\boldsymbol{\theta}}(\boldsymbol{x}|\boldsymbol{z}) + \log\mathcal{P}(\boldsymbol{z}) - \log q_{\boldsymbol{\theta}}(\boldsymbol{z}|\boldsymbol{x})\right]\;. \tag{24}$$

To connect the loss in Equation (24) to MIM, we first add the expectation of $\log\mathcal{P}(\boldsymbol{x})$, and scale the loss by a factor of $\frac{1}{2}$, to obtain

$$\mathbb{E}_{\boldsymbol{x}\sim\mathcal{P}(\boldsymbol{x}),\boldsymbol{z}\sim q_{\boldsymbol{\theta}}(\boldsymbol{z}|\boldsymbol{x})}\left[-\frac{1}{2}\left(\log(p_{\boldsymbol{\theta}}(\boldsymbol{x}|\boldsymbol{z})\mathcal{P}(\boldsymbol{z})) + \log(q_{\boldsymbol{\theta}}(\boldsymbol{z}|\boldsymbol{x})\mathcal{P}(\boldsymbol{x}))\right) + \log\mathcal{P}(\boldsymbol{x}) + \log q_{\boldsymbol{\theta}}(\boldsymbol{z}|\boldsymbol{x})\right] \tag{25}$$

where $\mathcal{P}(\boldsymbol{x})$ is the data distribution, which is assumed to be independent of model parameters $\theta$ and to exist almost everywhere (i.e., complementing $\mathcal{P}(\boldsymbol{z})$). Importantly, because $\mathcal{P}(\boldsymbol{x})$ does not depend on $\boldsymbol{\theta}$, the gradients of Eqs. (24) and (25) are identical up to a multiple of $\frac{1}{2}$, so they share the same stationary points.

Combining IID samples from the data distribution, $\boldsymbol{x}^i \sim \mathcal{P}(\boldsymbol{x})$, with samples from the corresponding variational posterior, $\boldsymbol{z}^i \sim q_{\boldsymbol{\theta}}(\boldsymbol{z}|\boldsymbol{x}^i)$, we obtain a joint sampling distribution; i.e.,

$$\mathcal{M}_{\mathcal{S}}^{\text{VAE}}(\boldsymbol{x},\boldsymbol{z}) \;=\; \mathcal{P}(\boldsymbol{x})\,q_{\boldsymbol{\theta}}(\boldsymbol{z}|\boldsymbol{x})$$

where $\mathcal{M}_{\mathcal{S}}^{\text{VAE}}$ comprises the encoding distribution in $\mathcal{M}_{\mathcal{S}}$. With it one can then rewrite the objective in Equation (25) in terms of the cross-entropy between $\mathcal{M}_{\mathcal{S}}^{\text{VAE}}$ and the parametric encoding and decoding distributions; i.e.,

$$\frac{1}{2}\big(H(\mathcal{M}_{\mathcal{S}}^{\text{VAE}}, p_{\boldsymbol{\theta}}(\boldsymbol{x}|\boldsymbol{z})\,\mathcal{P}(\boldsymbol{z})) + H(\mathcal{M}_{\mathcal{S}}^{\text{VAE}}, q_{\boldsymbol{\theta}}(\boldsymbol{z}|\boldsymbol{x})\,\mathcal{P}(\boldsymbol{x}))\big) +$$
$$- H_{\mathcal{M}_{\mathcal{S}}^{\text{VAE}}}(\boldsymbol{x}) - H_{\mathcal{M}_{\mathcal{S}}^{\text{VAE}}}(\boldsymbol{z}) + I_{\mathcal{M}_{\mathcal{S}}^{\text{VAE}}}(\boldsymbol{x};\boldsymbol{z})\;. \tag{26}$$

The sum of the last three terms in Equation (26) is the negative joint entropy $-H(\boldsymbol{z},\boldsymbol{x})$ under the sample distribution $\mathcal{M}_{\mathcal{S}}^{\text{VAE}}$.

Equations (1) and (26), the VAE objective and VAE as regularized cross entropy objective respectively, define equivalent optimization problems, under the assumption that $\mathcal{P}(\boldsymbol{x})$ and samples $\boldsymbol{x}\sim\mathcal{P}(\boldsymbol{x})$ do not depend on the parameters $\boldsymbol{\theta}$, and that the optimization is gradient-based. Formally, the VAE objectives (1) and (26) are equivalent up to a scalar multiple of $\frac{1}{2}$ and an additive constant, namely, $H_{\mathcal{M}_{\mathcal{S}}^{\text{VAE}}}(\boldsymbol{x})$.

Equation (26) is the average of two cross-entropy objectives ( *i.e.*, between sample distribution $\mathcal{M}_{\mathcal{S}}^{\text{VAE}}$ and the model decoding and encoding distributions, respectively), along with a joint entropy term (*i.e.*, last three terms), which can be viewed as a regularizer that encourages a reduction in mutual information and increased entropy in $\boldsymbol{z}$ and $\boldsymbol{x}$. We note that Equation (26) is similar to the MIM objective in Equation (8), but with a different sample distribution, where the priors are defined to be the anchors, and with an additional regularizer. In other words, Equation (26) suggests that VAE learning implicitly lowers mutual information. This runs contrary to the goal of learning useful latent representations, and we posit that it is an underlying root cause for *posterior collapse*, wherein the trained model show low mutual information which can be manifested as an encoder which matches the prior, and thus provides weak information about the latent state (e.g., see (Chen et al., 2016b) and others). We point the reader to Section E.1 for empirical evidence for the use of a joint entropy as a mutual information regularizer.

## E  ADDITIONAL EXPERIMENTS

Here we provide additional experiments that further explore the characteristics of MIM learning.

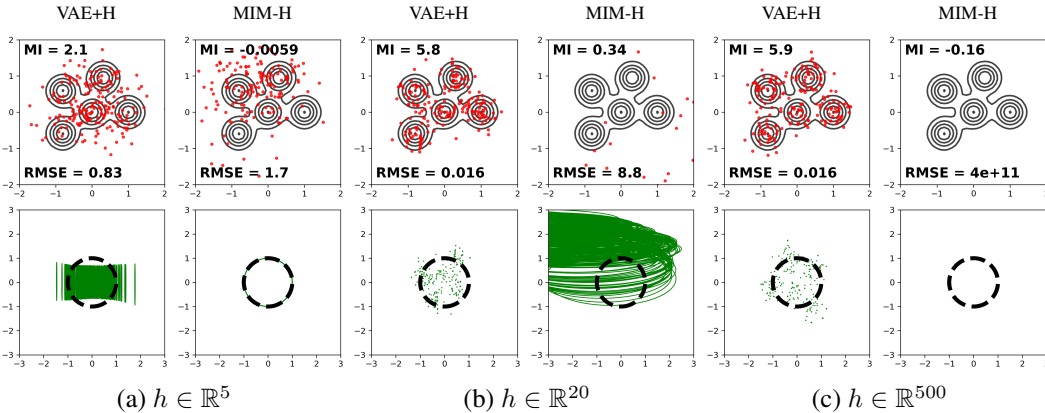

(a) $h \in \mathbb{R}^5$      (b) $h \in \mathbb{R}^{20}$      (c) $h \in \mathbb{R}^{500}$

Figure 7: Effects of entropy as a mutual information regularizer in 2D $\boldsymbol{x}$ and 2D $\boldsymbol{z}$ synthetic problem. VAE and MIM models with 2D inputs, a 2D latent space, and 5, 20 and 500 hidden units. Top row: Black contours depict level sets of $\mathcal{P}(\boldsymbol{x})$; red dots are reconstructed test points. Bottom row: Green contours are one standard deviation ellipses of $q_{\boldsymbol{\theta}}(\boldsymbol{z}|\boldsymbol{x})$ for test points. Dashed black circles depict one standard deviation of $\mathcal{P}(\boldsymbol{z})$. Here we added $H_{q_{\boldsymbol{\theta}}}(\boldsymbol{x}, \boldsymbol{z})$ to VAE loss, and subtracted $H_{\mathcal{M}_{\mathcal{S}}}(\boldsymbol{x}, \boldsymbol{z})$ from MIM loss, in order to demonstrate the effect of entropy on mutual information. Posterior collapse in VAE is mitigated following the increased mutual information. MIM, on the other hand, demonstrates a severe posterior collapse as a result of the reduced mutual information (*i.e.*, posterior matches prior over $\boldsymbol{z}$ almost perfectly). (see inset quantities).

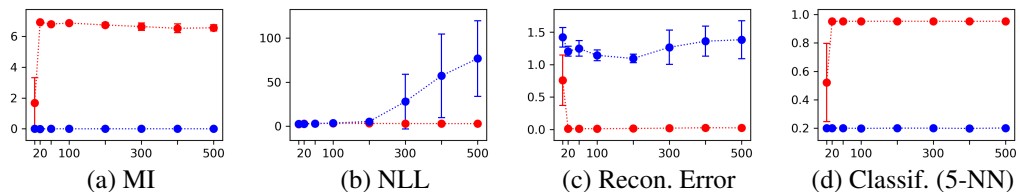

(a) MI      (b) NLL      (c) Recon. Error      (d) Classif. (5-NN)

Figure 8: Effects of entropy as a mutual information regularizer in 2D $\boldsymbol{x}$ and 2D $\boldsymbol{z}$ synthetic problem. Test performance for modified MIM (blue) and modified VAE (red) for the 2D GMM data with (cf. Fig. 7), all as functions of the number of hidden units (on x-axis). Each plot shows the mean and standard deviation of 10 experiments. Adding encoding entropy regularizer to VAE loss leads to high mutual information (*i.e.*, prevent posterior collapse), low reconstruction error, and better classification accuracy. Subtracting sample entropy regularizer from MIM loss results in almost zero mutual information (severe collapse), which leads to poor reconstruction error and classification accuracy.

### E.1 ENTROPY AS MUTUAL INFORMATION REGULARIZER

Here we examine the use of entropy as a mutual information regularizer. We repeat the experiment in Section 5.1 with added entropy regularizer. Figure 7 depicts the effects of an added $H_{q_{\boldsymbol{\theta}}}(\boldsymbol{x}, \boldsymbol{z})$ to the VAE loss, and a subtracted $H_{\mathcal{M}_{\mathcal{S}}}(\boldsymbol{x}, \boldsymbol{z})$ from MIM . The corresponding quantitative values are presented in Figure 8. Adding the entropy regularizer leads to increased the mutual information, and subtracting it results in a strong posterior collapse, which in turn is reflected in the reconstruction quality. While such an experiment does not represent a valid probabilistic model, it supports our use of entropy as a regularizer (cf. Eq. (5)) for JSD in order to define a consistent model with high mutual information.

### E.2 CONSISTENCY REGULARIZER IN $\mathcal{L}_{\text{MIM}}$

Here we explore properties of models for 1D $\boldsymbol{x}$ and $\boldsymbol{z}$, learned with $\mathcal{L}_{\text{MIM}}$ and $\mathcal{L}_{\text{CE}}$, the difference being the model consistency regularizer $\text{R}_{\text{MIM}}(\boldsymbol{\theta})$. All model priors and conditional likelihoods $(q_{\boldsymbol{\theta}}(\boldsymbol{x}), q_{\boldsymbol{\theta}}(\boldsymbol{z}|\boldsymbol{x}), p_{\boldsymbol{\theta}}(\boldsymbol{z}), p_{\boldsymbol{\theta}}(\boldsymbol{x}|\boldsymbol{z}))$ are parameterized as 10-component Gaussian mixture models, and

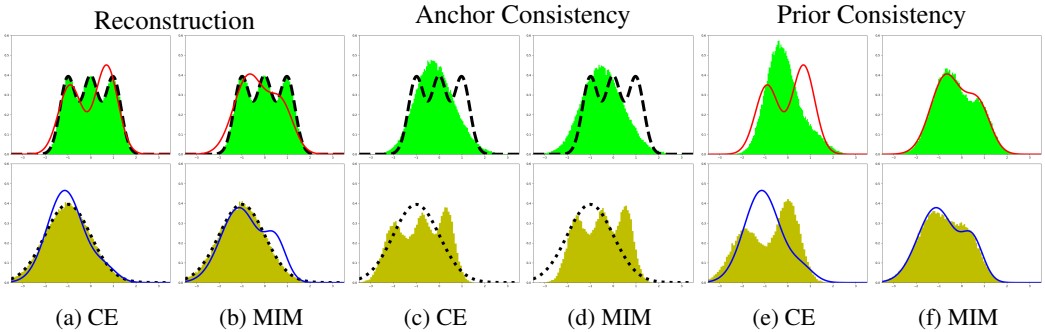

|                  |                  |                  |
| :--------------: | :--------------: | :--------------: |
| Reconstruction   | Anchor Consistency | Prior Consistency |

(a) CE     (b) MIM     (c) CE     (d) MIM     (e) CE     (f) MIM

Figure 9: We explore the influence of consistency regularizer $R_{\theta}$. CE and MIM indicate the loss, $\mathcal{L}_{CE}$ or $\mathcal{L}_{MIM}$ (the regularized objective), respectively. Top row shows anchor $\mathcal{P}(x)$ (dashed), prior $q_{\theta}(x)$ (red), and reconstruction distribution $x_i \sim \mathcal{P}(x) \to z_i \sim q_{\theta}(z|x_i) \to x_i' \sim p_{\theta}(x|z_i)$ (green). Bottom row mirrors the top row, with anchor $\mathcal{P}(z)$ (dotted), prior $p_{\theta}(z)$ (blue), and reconstruction distribution $z_i \sim \mathcal{P}(z) \to x_i \sim p_{\theta}(x|z_i) \to z_i' \sim q_{\theta}(z|x)$ (yellow). In (c-d) the reconstruction is replaced with decoding from anchors $z_i \sim \mathcal{P}(z) \to x_i' \sim p_{\theta}(x|z_i)$ (green), and encoding $x_i \sim \mathcal{P}(x) \to z_i' \sim q_{\theta}(z|x)$ (yellow). In (e-f) the reconstruction is replaced with decoding from priors $z_i \sim p_{\theta}(z) \to x_i' \sim p_{\theta}(x|z_i)$ (green), and encoding $x_i \sim q_{\theta}(x) \to z_i' \sim q_{\theta}(z|x)$ (yellow). While both models offers similar reconstruction (a-b), and similar consistency w.r.t. the anchors (c-d), only MIM finds a consistent model (e-f). See text for details.

optimized during training. Means and variances for the conditional distributions were regressed with 2 fully connected layers ($h \in \mathbb{R}^{10}$) and a swish activation function Ramachandran et al. (2018).

Top and bottom rows in Fig. 9 depict distributions in observations and latent spaces respectively. Dashed black curves are anchors, $\mathcal{P}(x)$ on top, and $\mathcal{P}(z)$ below (GMMs with up to 3 components). Learned model priors, $q_{\theta}(x)$ and $p_{\theta}(z)$, are depicted as red (top) and blue (bottom) curves.

Green histograms in Fig. 9(a,b) depict reconstruction distributions, computed by passing fair samples from $\mathcal{P}(x)$ through the encoder to $z$ and then back through the decoder to $x$. Similarly the yellow histograms shows samples from $\mathcal{P}(z)$ passed through the decoder and then back to the latent space. For both losses these reconstruction histograms match the anchor priors well. In contrast, only the priors that were learned with $\mathcal{L}_{CE}$ loss approximates the anchor well, while the $\mathcal{L}_{MIM}$ priors do not. To better understand that, we consider two generative procedures: sampling from the anchors, and sampling from the priors.

Anchor consistency is depicted in Fig. 9(c,d), where Green histograms are marginal distributions over $x$ from the anchored decoder (i.e., samples from $\mathcal{P}(z)p_{\theta}(x|z)$ ). Yellow are marginals over $z$ from the anchored encoders $\mathcal{P}(x)q_{\theta}(z|x)$. One can see that both losses results in similar quality of matching the corresponding opposite anchors.

Priors consistency is depicted in Fig. 9(e,f), where Green histograms are marginal distributions over $x$ from the model decoder $p_{\theta}(z)p_{\theta}(x|z)$. Yellow depicts marginals over $z$ from the model encoder $q_{\theta}(x)q_{\theta}(z|x)$. Importantly, with $\mathcal{L}_{MIM}$ the encoder and decoder are consistent; i.e., $q_{\theta}(x)$ (red curve) matches the decoder marginal, while $p_{\theta}(z)$ (blue) matches the encoder marginal. The model trained with $\mathcal{L}_{CE}$ (*i.e.*, without consistency prior) fails to learn a consistent encoder-decoder pair. We note that in practice, with expressive enough priors, $\mathcal{L}_{MIM}$ will be a tight bound for $\mathcal{L}_{CE}$.

### E.3 PARAMETERIZING THE PRIORS

Here we explore the effect of parameterizing the latent and observed priors. A fundamental idea in MIM is the concept of a single model, $\mathcal{M}_{\theta}$. As such, parameterizing a prior increases the global expressiveness of the model $\mathcal{M}_{\theta}$. Fig. 10 depicts the utilization of the added expressiveness in order to increase the consistency between the encoding and decoding model distribution, in addition to the consistency of $\mathcal{M}_{\theta}$ with $\mathcal{M}_{\mathcal{S}}$.

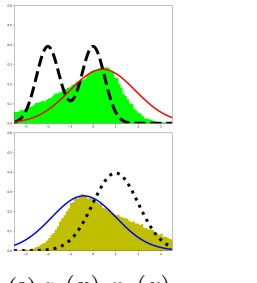 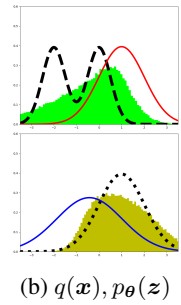 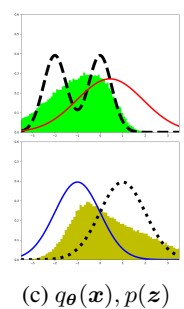 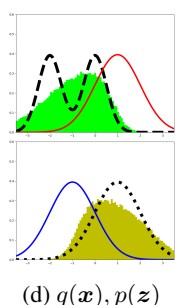

(a) $q_{\boldsymbol{\theta}}(\boldsymbol{x}), p_{\boldsymbol{\theta}}(\boldsymbol{z})$  (b) $q(\boldsymbol{x}), p_{\boldsymbol{\theta}}(\boldsymbol{z})$  (c) $q_{\boldsymbol{\theta}}(\boldsymbol{x}), p(\boldsymbol{z})$  (d) $q(\boldsymbol{x}), p(\boldsymbol{z})$

Figure 10: MIM prior expressiveness. In this experiment we explore the effect of learning a prior, where the priors $q(\boldsymbol{x})$ and $p(\boldsymbol{z})$ are normal Gaussian distributions. Top row shows anchor $\mathcal{P}(\boldsymbol{x})$ (dashed), prior $q_{\boldsymbol{\theta}}(\boldsymbol{x})$ (red), and decoding distribution $\boldsymbol{z}_i \sim p_{\boldsymbol{\theta}}(\boldsymbol{z}) \to x'_i \sim p_{\boldsymbol{\theta}}(\boldsymbol{x}|\boldsymbol{z}_i)$ (green). Bottom row mirrors the top row, with anchor $\mathcal{P}(\boldsymbol{z})$ (dotted), prior $p_{\boldsymbol{\theta}}(\boldsymbol{z})$ (blue), and encoding distribution $\boldsymbol{x}_i \sim q_{\boldsymbol{\theta}}(\boldsymbol{x}) \to z'_i \sim q_{\boldsymbol{\theta}}(\boldsymbol{z}|\boldsymbol{x})$ (yellow). As can be seen, parameterizing priors affects all learned distributions, supporting the notion of optimization of a single model $\mathcal{M}_{\boldsymbol{\theta}}$. We point that (a) demonstrates the best consistency between the priors and corresponding generated samples, following the additional expressiveness.

### E.4 EFFECT OF CONSISTENCY REGULARIZER ON OPTIMIZATION

Here we explore whether a learned model with consistent encoding-decoding distributions (*i.e.*, trained with $\mathcal{L}_{\mathrm{MIM}}$) also constitutes an optimal solution of a CE objective (*i.e.*, trained with $\mathcal{L}_{\mathrm{CE}}$). Results are depicted in Fig. 11. In order to distinguish between the effects of the optimization from the consistency regularizer we initialize a MIM model by pre-training it with $\mathcal{L}_{\mathrm{CE}}$ loss followed by $\mathcal{L}_{\mathrm{MIM}}$ training in Fig. 11(i), and vice verse in Fig. 11(ii). (a-b,e-f) All trained models in Fig. 11 exhibit similarly good reconstruction (green matches dashed black). (c-d,g-h) However, only models that were trained with $\mathcal{L}_{\mathrm{MIM}}$ exhibit encoding-decoding consistency (green matches red, yellow matches blue). While it is clear that the optimization plays an important role (*i.e.*, different initialization leads to different local optimum), it is also clear that encoding-decoding consistency is not necessarily an optimum of $\mathcal{L}_{\mathrm{CE}}$, as depicted in a non-consistent model (h) which was initialized with a consistent model (g). Not surprisingly, without the consistency regularizer training with $\mathcal{L}_{\mathrm{CE}}$ results in better fit of priors to anchors (f) as it is utilizing the expressiveness of the parametric priors in matching the sample distribution.

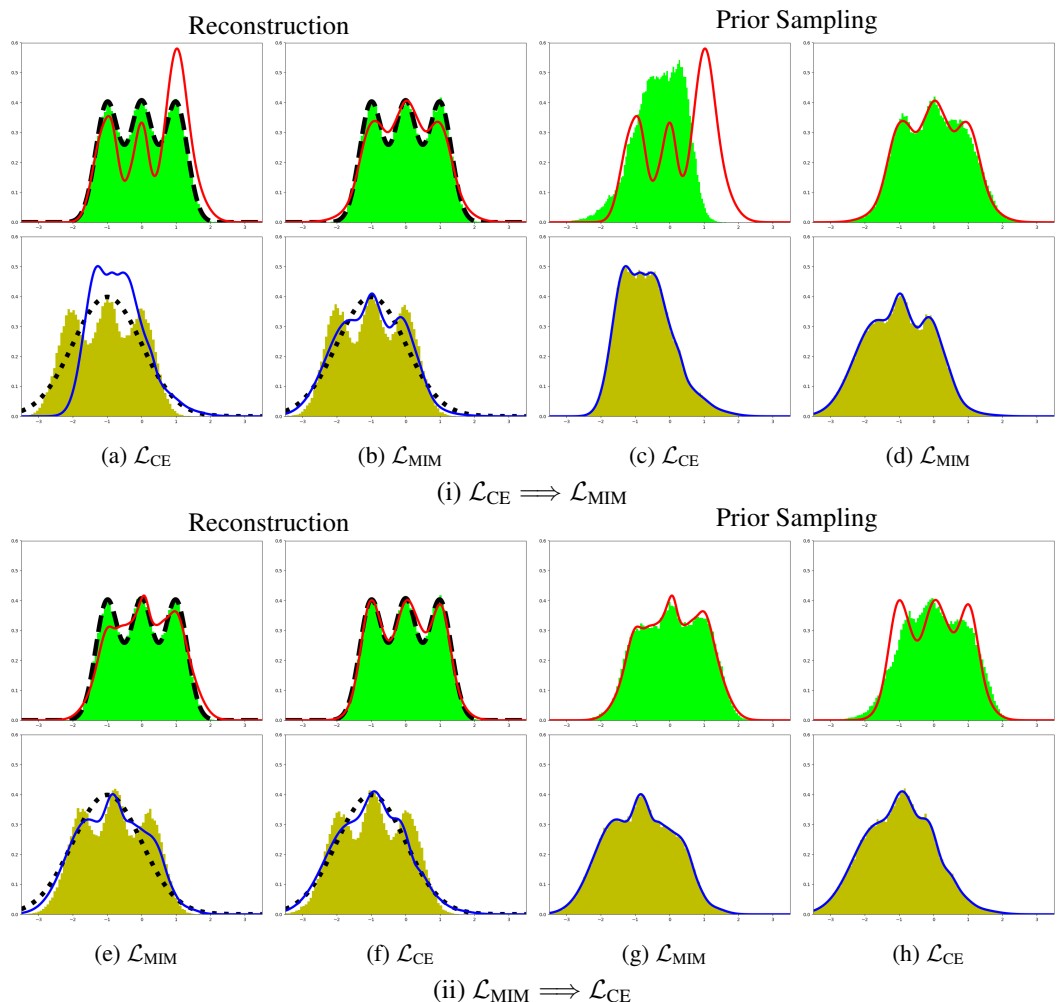

Figure 11: Effects of MIM consistency regularizer and optimization on encoding-decoding consistency. (i) and (ii) differ in initialization order. Odd rows: anchor $\mathcal{P}(\boldsymbol{x})$ (dashed), prior $q_{\boldsymbol{\theta}}(\boldsymbol{x})$ (red). Even rows: anchor $\mathcal{P}(\boldsymbol{z})$ (dotted), prior $p_{\boldsymbol{\theta}}(\boldsymbol{z})$ (blue). (a-b,e-f) Reconstruction $\boldsymbol{x}_i \sim \mathcal{P}(\boldsymbol{x}) \rightarrow \boldsymbol{z}_i \sim q_{\boldsymbol{\theta}}(\boldsymbol{z}|\boldsymbol{x}_i) \rightarrow x'_i \sim p_{\boldsymbol{\theta}}(\boldsymbol{x}|\boldsymbol{z}_i)$ ($x'_i$ green, $\boldsymbol{z}_i$ yellow). (c-d,g-h) Prior decoding $\boldsymbol{z}_i \sim p_{\boldsymbol{\theta}}(\boldsymbol{z}) \rightarrow x'_i \sim p_{\boldsymbol{\theta}}(\boldsymbol{x}|\boldsymbol{z}_i)$ (green), and prior encoding $\boldsymbol{x}_i \sim q_{\boldsymbol{\theta}}(\boldsymbol{x}) \rightarrow \boldsymbol{z}_i \sim q_{\boldsymbol{\theta}}(\boldsymbol{z}|\boldsymbol{x}_i)$ (yellow). See text for details.

# F ADDITIONAL RESULTS

Here we provide additional visualization of various MIM and VAE models which were not included in the main body of the paper.

## F.1 RECONSTRUCTION AND SAMPLES FOR MIM AND A-MIM

In what follows we show samples and reconstruction for MIM (*i.e.*, with convHVAE architecture), and A-MIM (*i.e.*, with PixelHVAE architecture). We demonstrate, again, that a powerful enough encoder allows for generation of samples which are comparable to VAE samples.

Standard Prior                    VampPrior Prior

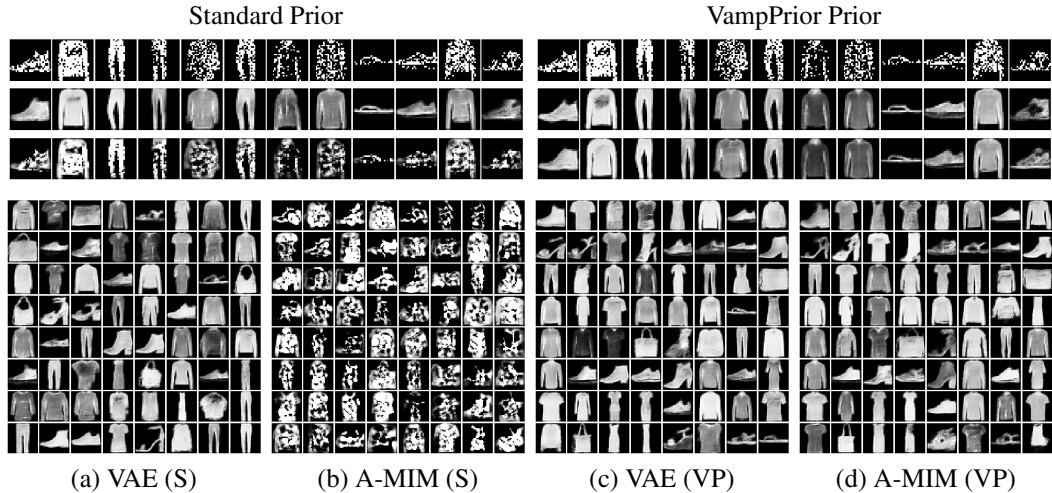

(a) VAE (S)          (b) A-MIM (S)          (c) VAE (VP)          (d) A-MIM (VP)

Figure 12: MIM and VAE learning with PixelHVAE for Fashion MNIST. The top three rows (from top to bottom) are test data samples, VAE reconstruction, A-MIM reconstruction. Bottom: random samples from VAE and MIM. (c-d) We initialized all pseudo-inputs with training samples, and used the same random seed for both models. As a result the samples order is similar.

Standard Prior                    VampPrior Prior

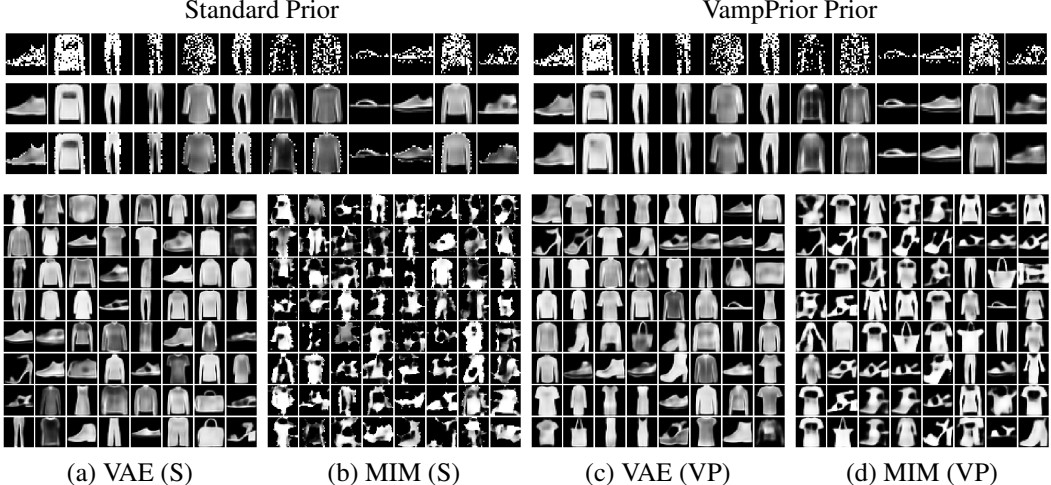

(a) VAE (S)          (b) MIM (S)          (c) VAE (VP)          (d) MIM (VP)

Figure 13: MIM and VAE learning with convHVAE for Fashion MNIST. The top three rows (from top to bottom) are test data samples, VAE reconstruction, MIM reconstruction. Bottom: random samples from VAE and MIM.

Standard Prior          VampPrior Prior

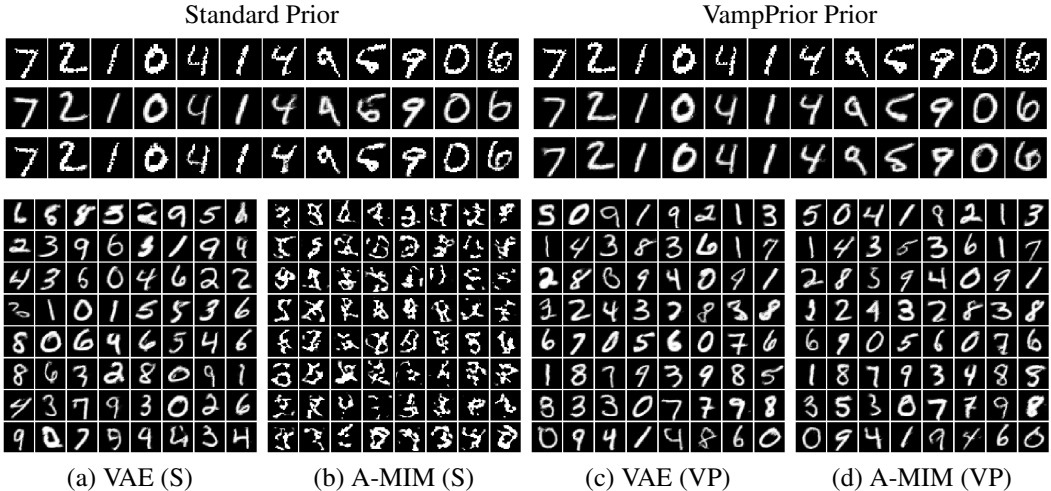

(a) VAE (S)     (b) A-MIM (S)     (c) VAE (VP)     (d) A-MIM (VP)

Figure 14: MIM and VAE learning with PixelHVAE for MNIST. Top three rows are test data samples, followed by VAE and A-MIM reconstructions. Bottom: random samples from VAE and MIM. (c-d) We initialized all pseudo-inputs with training samples, and used the same random seed for both models. As a result the samples order is similar.

Standard Prior          VampPrior Prior

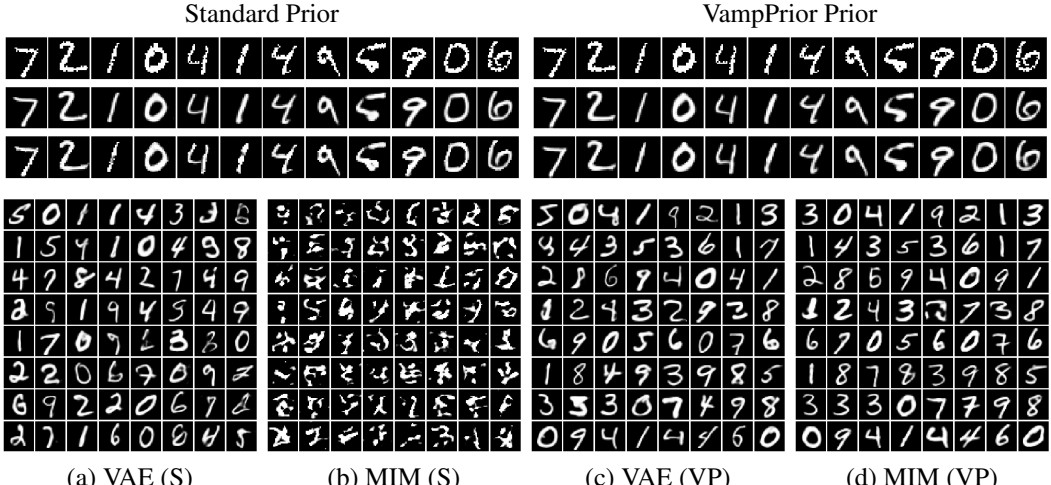

(a) VAE (S)     (b) MIM (S)     (c) VAE (VP)     (d) MIM (VP)

Figure 15: MIM and VAE learning with convHVAE for MNIST. Top three rows are test data samples, followed by VAE and MIM reconstructions. Bottom: random samples from VAE and MIM.

Standard Prior                    VampPrior Prior

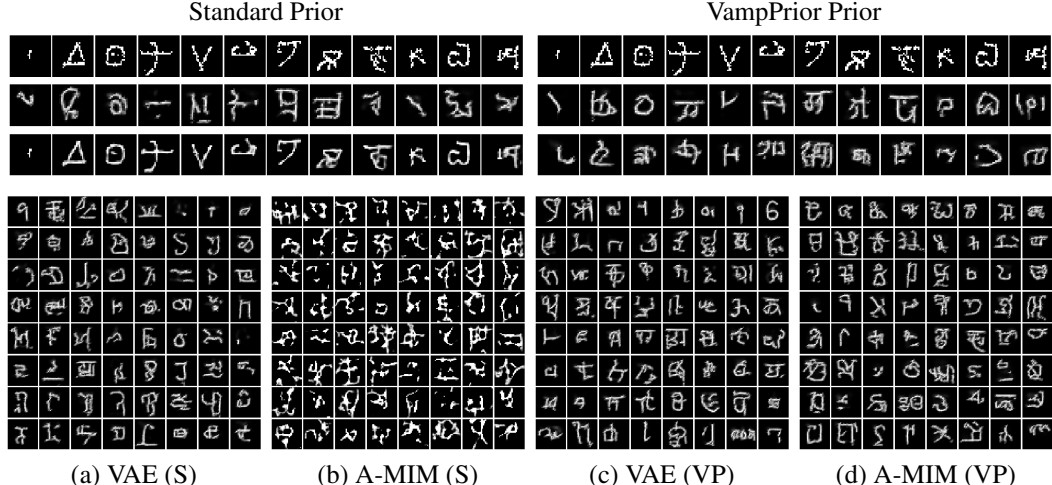

(a) VAE (S)          (b) A-MIM (S)          (c) VAE (VP)          (d) A-MIM (VP)

Figure 16: MIM and VAE learning with PixelHVAE for Omniglot. Top three rows are test data samples, followed by VAE and A-MIM reconstructions. Bottom: random samples from VAE and MIM.

Standard Prior                    VampPrior Prior

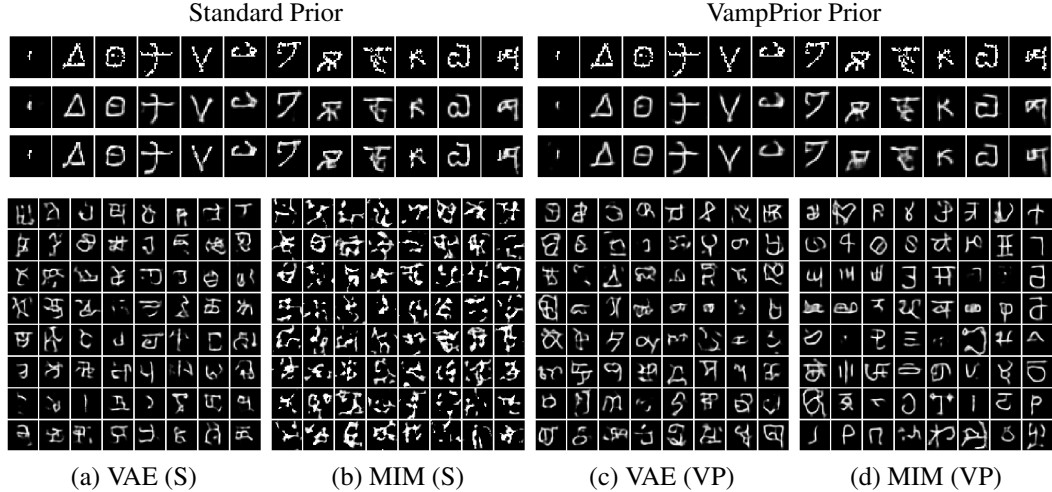

(a) VAE (S)          (b) MIM (S)          (c) VAE (VP)          (d) MIM (VP)

Figure 17: MIM and VAE learning with convHVAE for Omniglot. Top three rows are test data samples, followed by VAE and MIM reconstructions. Bottom: random samples from VAE and MIM.

## F.2   LATENT EMBEDDINGS FOR MIM AND A-MIM

In what follows we show additional t-SNE visualization of unsupervised clustering in the latent representation for MIM (*i.e.*, with convHVAE architecture), and A-MIM (*i.e.*, with PixelHVAE architecture).

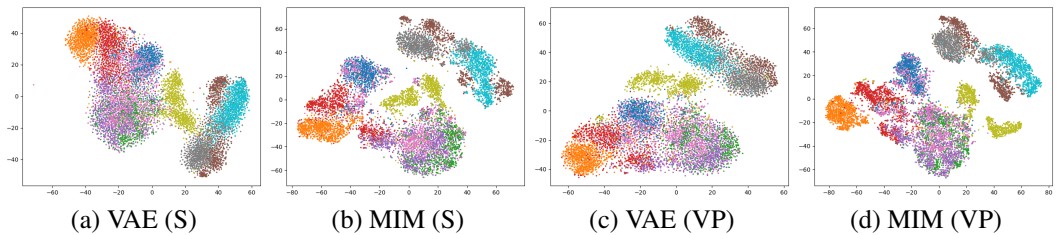

(a) VAE (S)     (b) MIM (S)     (c) VAE (VP)     (d) MIM (VP)

Figure 18: MIM and VAE $z$ embedding for Fashion MNIST with convHVAE architecture.

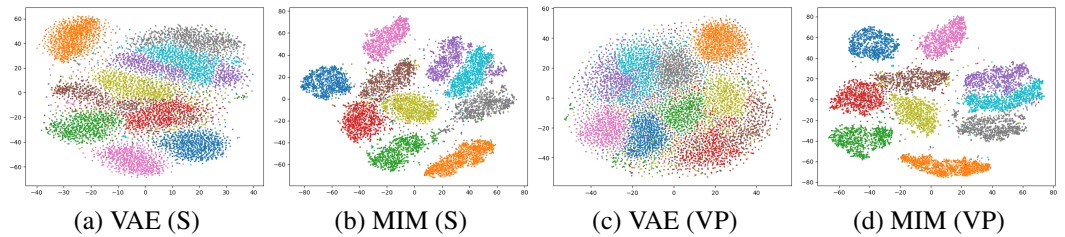

(a) VAE (S)     (b) MIM (S)     (c) VAE (VP)     (d) MIM (VP)

Figure 19: MIM and VAE $z$ embedding for MNIST with convHVAE architecture.

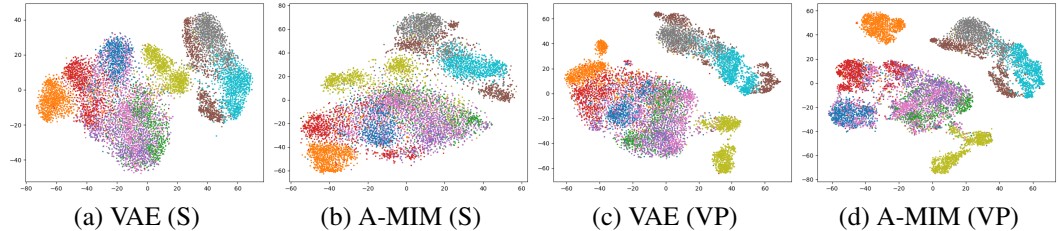

(a) VAE (S)     (b) A-MIM (S)     (c) VAE (VP)     (d) A-MIM (VP)

Figure 20: A-MIM and VAE $z$ embedding for Fashion MNIST with PixelHVAE architecture.

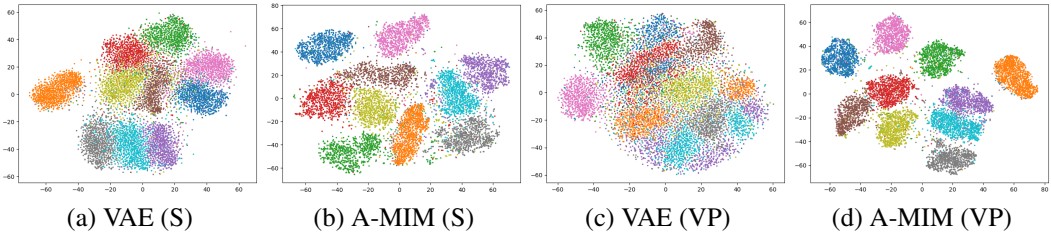

(a) VAE (S)     (b) A-MIM (S)     (c) VAE (VP)     (d) A-MIM (VP)

Figure 21: A-MIM and VAE $z$ embedding for MNIST with PixelHVAE architecture.

