# OpenReview forum: "MIM: Mutual Information Machine"
_ICLR.cc/2020/Conference — Reject_

### Official Review · AnonReviewer3 · 2019-10-16
**Official Blind Review #3**

**Rating:** 1

**Review:**

The authors present the Mutual Information Machine (MIM), which is essentially a  latent variable model learned via Adversarially Learned Inference with regularizations that prefer learning representations with high mutual information. While the method is interesting, it does not seem to be well justified in representation learning, and the empirical results are not well-justified due to the use of weak baselines.

Pros:

The idea of formulating the L_MIM objective appears interesting, which uses the mixture of P_\theta and Q_\theta and minimizes an upper bound to H(x) + H(z) - I(x, z), so it is effectively minimizing a lower bound to I(x, z), under the proposed symmetric distribution.

Cons:

Symmetry argument:
	- Prior work on maximizing mutual information for latent variables focus on the MI as evaluated in the "encoder distribution", whereas in this paper the argument is focused on symmetry, i.e. the MI estimated is based on \mathcal{M}_s.
	- However, for practical purposes, it is unclear how this is superior than the "encoder distribution" paradigm. Eventually, we wish to use the z for some downstream tasks, which requires us to obtain z from some distribution (e.g. the encoder q(z|x)) and then run classification on this z.
	- Unless we sample z | x from the conditional distribution of \mathcal{M}_s as proposed by the authors, it is unclear why we need this symmetry in representation learning; it seems slower to sample from the MIM distribution though (see motivation for A-MIM).
	- The authors did not demonstrate results that compare A-MIM and MIM on the same architecture.

Failure to discuss certain related literature, such as Alemi et al. 2017 and Zhao et al. 2018. These are also "strongly related to VAEs" and related to mutual information objectives with latent variables.
	- Alemi et al. 2017 discussed this in the context of "rate discussion trade-off"
	- Zhao et al. 2018 proposed objectives to find specific trade-offs under constraints
In fact, Zhao et al. 2018 considered the MIs in both P_theta and Q_theta, and one should be able to obtain a "symmetric" MI by adding these two Mis uniformly (although again, I am not sure how this is useful for downstream applications on the model).

Using VAE as a baseline comparison.
	- It is well known that the VAE objective does not encourage maximizing mutual information, and various existing works have already been proposed to address this problem. Outperforming VAE in terms of MI is hence not very surprising.
	- Methods that encourage mutual information maximization under the latent variable generative modeling framework would be more suited for comparison. How does MIM compare with the methods introduced in Chen et al. 2016, Alemi et al. 2017 and Zhao et al. 2018?
	- In the context of ALI and BiGAN, one could also apply other approaches like MINE / InfoMAX to maximize MI. Again, there is no comparison with these approaches.
	- Alemi and Zhao both showed that there is a trade-off between consistency and mutual information under the same architecture (using their objectives). It seems that what MIM empirically does is also to sacrifice NLL for better MI, and not clear whether MIM gives comparable or better trade-offs.

The paper would be more convincing if the authors empirically compare with other sensible baselines and justify why we should use MIM as opposed to A-MIM in representation learning.

[relevant papers]
Chen et al. 2016 Lossy VAEs
Alemi et al. 2017 Fixing a broken ELBO
Zhao et al. 2018 A Lagrangian Perspective to Latent Variable Generative Models


**Experience Assessment:**

I have published in this field for several years.

**Review Assessment: Checking Correctness Of Derivations And Theory:**

I assessed the sensibility of the derivations and theory.

**Review Assessment: Checking Correctness Of Experiments:**

I carefully checked the experiments.

**Review Assessment: Thoroughness In Paper Reading:**

I read the paper at least twice and used my best judgement in assessing the paper.

---

> ### Author Response · Authors · 2019-11-11
> **Thank you very much for your detailed review. Please find our reply below.**
>
> Thank you very much for your detailed review. We were not aware of Zhao et al. 2018, and we find the paper both interesting and useful. Likewise, we will cite/discuss Chen et al. and Alemi et al. Please see below for responses to your comments.
>
> MIM vs ALI: We agree with the similarity of MIM and ALI. However,  since MIM is using a variational bound and not adversarial learning, MIM learns a probability density estimator over the joint distribution (which is suitable for downstream tasks) whereas ALI learns only the sampling component of said density estimator. One of the major strengths of MIM vs ALI (and similar approaches) is that we can train MIM by gradient descent with reparameterization as opposed to adversarial learning. Note that Zhao et al (2018) iin Sec. 2  states that this joint divergence optimization typically belongs to the hardest category of optimization problems, usually necessitating adversarial learning. As a consequence, we believe our approach offers something new and significant to the community.
>
> A-MIM vs MIM: Regarding A-MIM vs MIM, we chose to omit a direct comparison for clarity and to save space, but we can include it.  Using MIM adds on average 1-2% in classification accuracy, without any significant effect on NLL (i.e., it is a better representation for “downstream tasks”).
>
> Comparison to VAE: We chose to focus on comparisons with VAE as it is the most similar learning framework (i.e., MIM and VAE both optimize a divergence over the joint distribution D(q(x, z) || p(x, z)) using variational inference). Our goal here was to demonstrate that an alternative formulation (i.e., MIM)  can offer the benefits of VAE (NLL, sampling), with superior representation. We also acknowledge that additional comparisons to other MI-related methods would be beneficial. We will include comparisons to other frameworks that encourage high mutual information in future work that focuses on representation learning.
>
> The desire for symmetry: The same argument against symmetry could be made for ALI/BiGAN/AS-VAE(Pu et al., 2017)/sVAE (chen et al., 2018) and any other symmetric encoder/decoder paradigm. Our motivations are similar to these other papers. In particular, see sVAE (“Symmetric Variational Autoencoder and Connections to Adversarial Learning” Chen et al., 2018)  section 2.2 “limitations of the VAE” for a detailed explanation. Furthermore, in our case, when symmetry is achieved, then the joint entropy objective RH (eq. 5) can be shown to reduce to exactly mutual information (see our comments above to Jianlin Su for more details). We will update the introduction to our paper to reflect the desire for symmetry in terms of the limitations of the VAE.  We will also include the ability to target mutual information via symmetry, in order to strengthen the motivation.
>
> Regarding the need to sample from the mixture distribution, we can actually sample from the encoding distribution instead. This is because we introduced the consistency regularizer R_MIM (Eq. 9) to learn a consistent encoding/decoding distribution.
> This allows one to sample the encoder for “downstream tasks” which is empirically consistent with the decoder (as demonstrated in the classification task), and as such provides a good approximation for fair samples from the mixture model (i.e., for an expressive enough architecture).
>
> We also note that when we estimated mutual information in our paper we did so using samples from the encoder alone, not from M_s.  This enables direct comparison with VAE, and indeed, if one is concerned mainly with representation learning then this can be considered a crucial quantity.  Nevertheless, with the divergence objective, as the model becomes more and more consistent the encoder and M_s become more and more similar.
>
> Comparisons to other approaches: We agree that further comparisons to other models would strengthen the paper. However, we note that unlike Alemi/Zhao, we have no hyper-parameters to controls the trade-offs (which would have to be determined empirically), nor is there any need for a difficult dual optimization. We empirically demonstrate that MIM formulation finds a solution with similar consistency to VAE (as measured by NLL/ELBO) and higher mutual information (as measured directly, or by reconstruction). We consider the lack of hyper-parameters as one of MIM’s advantages. For example, Zhao acknowledges the difficulty of finding the epsilon hyper-parameter (section 6), and resorts to a heuristic solution. It is likely that similar approaches to making the tradeoff more explicit could be derived with MIM, as it has for VAEs, but we think this is orthogonal to the purpose of this paper, which introduces and explores a new estimator, and therefore we argue that further exploration this area should be the subject of future work.

---

### Official Review · AnonReviewer2 · 2019-10-23
**Official Blind Review #2**

**Rating:** 1

**Review:**

When modeling a population distribution Pop(x) one typically minimizes the cross entropy

(1)  Theta^* = argmin_\Theta   H(Pop,P_\Theta)
                      = argmin_\Theta   E_{x \sim Pop}  -ln P_\Theta(x).

The ELBO used in the definition of a VAE provides an upper bound on this loss --- VAE is directly motivated by (1).

This paper defines a different loss function --- L_MIM.  However, I find the rather vaguely stated motivations for L_MIM uncompelling.  L_MIM does not bound (1) nor does it bound or estimate mutual information.  A lack of any compelling motivation could be overlooked if the empirical results where compelling.  But the empirical evaluation is very shallow with a focus on MNIST and similar data sets.

A technical comment is that I do not understand how z_i is generated in equation (12).  What distribution are the pairs (x_i,z_i) being drawn from?  I should note, however, that this technicality is not my main issue with the paper and will not influence my judgement.

Postscript: I have read the author's response and have not been swayed.  I still find the motivation obscure and the experiments weak.  The fact that other authors have done questionable things does not sway me.

**Experience Assessment:**

I have published one or two papers in this area.

**Review Assessment: Checking Correctness Of Derivations And Theory:**

I assessed the sensibility of the derivations and theory.

**Review Assessment: Checking Correctness Of Experiments:**

I assessed the sensibility of the experiments.

**Review Assessment: Thoroughness In Paper Reading:**

I made a quick assessment of this paper.

---

> ### Author Response · Authors · 2019-11-11
> **Thanks for your review. Please find our reply below.**
>
> Thank you for your review. We agree that the ELBO is meant to bound the maximum likelihood formulation given in (1). However, an equivalent formulation of VAEs (i.e., up to a constant) is to minimize DKL(q(x, z) || p(x, z)).  Indeed, the idea of minimizing a divergence between the two joint distributions is prevalent in the literature (e.g., [1,2,3]) and many of these can be seen to simultaneously optimize for mutual information (see [4]).  Our motivation is similar to these and other papers.
>
> The goal of this paper is not solely to bound or necessarily estimate mutual information, nor is it to optimize a bound on the log-likelihood of the data.  Rather, there is an established literature on methods seeking to optimize for high mutual information and a low divergence.  For example, see [2,5,6,7] and refer to the appendix in [4] for a unifying view of many such  methods under this objective. Specifically, our approach is most closely related to ALI/BiGAN [2,3] in seeking to minimize the Jensen-Shannon divergence between q(x,z) and p(x,z). We are also related to ALICE [7] in that we seek to encourage high mutual information.
>
> One of the differentiators between MIM and these works is that we do not need to resort to adversarial learning to optimize the loss, and instead derive an upper bound that allows direct gradient-based minimization.
>
> To clarify these issues and our motivation, we will revise the introduction of our paper to clarify the related work and our motivation for future readers.
>
> Regarding experiments, there is an established precedent in the literature for performing  experiments on well-known data and synthetic data to better understand properties of new models (eg, [8,9]). In our case, note that our experiments on MNIST, Fashion MNIST, and Omniglot follow the same protocol as the VampPrior paper [8] for baseline comparisons.
>
> Regarding your technical comment: Drawing samples from M_s entails drawing half the samples as x_i, z_i ~ P(x)q(z|x), and half the samples as x_i, z_i ~ p(x|z)P(z), where P(X) is data distribution (i.e., training set), and P(Z) is a Gaussian.  Please see Algorithm 1 in appendix for full details.
>
> [1] Symmetric Variational Autoencoder and Connections to Adversarial Learning (Chen et al. 2018)
> [2] ALI: Adversarially Learned Inference (Dumoulin et al. 2017)
> [3] BiGAN: Adversarial Feature Learning (Donahue et al. 2017)
> [4] A Lagrangian Perspective to Latent Variable Generative Models (Zhao et al. 2018)
> [5]  beta-vae: Learning basic visual concepts with a constrained variational framework (Higgins et al. 2016)
> [6] InfoGAN: Interpretable Representation Learning by Information Maximizing Generative Adversarial Nets (Chen et al. 2016)
> [7]  ALICE: Towards understanding adversarial learning for joint distribution matching (Li et al. 2017)
> [8] VAE with a VampPrior (Tomczak et al. 2018)
> [9] Adversarial Autoencoders (Makhzani et al. 2016)

---

### Official Review · AnonReviewer1 · 2019-10-26
**Official Blind Review #1**

**Rating:** 6

**Review:**

This paper defines a new learning objective of an autoencoder framework for learning joint distributions over observations and latent states, where the objective is the joint entropy of  M_s, an equally weighted mixture of the encoding and decoding distributions. Since it involves the marginal distribution which is intractable, it goes further to introduce an upper bound which is the cross-entropy between M_s and M_theta. To maintain consistency, it introduce an upper bound again, which is the average of cross entropy between the mixture distribution M_S and the model encoding and decoding distribution. Through comprehensive experiments, the effectiveness is validated.

I don't think Mutual Information Machine (MIM) is a proper name for this approach, since it intends to minimizing joint entropy M_s. Also the notation of cross-entropy H(p,q) is confusing, it's better to use CE(p,q) instead.

**Experience Assessment:**

I have read many papers in this area.

**Review Assessment: Checking Correctness Of Derivations And Theory:**

I carefully checked the derivations and theory.

**Review Assessment: Checking Correctness Of Experiments:**

I assessed the sensibility of the experiments.

**Review Assessment: Thoroughness In Paper Reading:**

I read the paper at least twice and used my best judgement in assessing the paper.

---

> ### Author Response · Authors · 2019-11-11
> **Thanks for your review. We like your notational suggestions.**
>
> Thank you for taking the time to review the paper. We like your notational suggestion for CE(p,q), and we will adopt it.
> In terms of the paper title, both symmetry and mutual information play important roles in our formulation. We also considered “Symmetric Variational Autoencoders” (SymVAE), and if you believe this is a better name, please let us know.

---

### Author Response · Authors · 2019-10-02
**Typo in stdev value in Table 1b.**

Should be "well less than 0.01", instead of "well less than 0.1"

---

### Author Response · Authors · 2019-10-03
**Typon in Eq. (12)**

The annealing of the loss has a typo, and should be

L_{MIM} ( \theta ; \{ x_i, z_i  \}_{i=1}^{N}, \beta  ) =
 -\frac{1}{2N} \sum_{i=1}^{N} \log (p_{\theta}(x_i|z_i) p_{\theta}(z_i)  ) + \beta \log ( q_{\theta} (z_i|x_i) \log q_{\theta}(x_i) )

or in short:

L_MIM = 1/(2N) ∑ log p(xi, zi) + β log q(xi,zi)

which encourages better sampling quality from the trained MIM model.

---

### Public Comment · ~Jianlin_Su1 · 2019-10-16
**unreasonable substitute**

I think replacing I(X,Y) with H(x, y) is unreasonable. H(x,y) = H(x) + H(y) - I(x,y), so to minimize H(x,y) may not maximize I(x,y) but may minimize H(x) + H(y).

Meanwhile, why the weight of regularizer H(x,y) is set to 1? Just for cobbling together a H(M(x))?

---

> ### Author Response · Authors · 2019-10-17
> **Thank you for your comments. See our reply below.**
>
> Thanks for your comments.  See below short and a more elaborated reply.
>
> * Comment 1 (short): Indeed, there might exist a pathological solution where H(x) + H(z) are minimized while I(x,z) is not maximized. However, we do show empirically that mutual information is increased with MIM, and an ablation (Appendix E.1) that shows how the joint entropy definitely affects the mutual information in our 2D example.
>
> * Comment 2: Choosing the weight of H(x,z) to be 1 allows for a variational upper bound of cross entropy over a mixture model M(x,z). This also yields a simple, interpretable objective,
> without the need for another hyper-parameter that, presumably, one would need to determine empirically.
> Using a different weight will lead to a different model and will require a different tractable variational bound.
>
> * Comment 1 (long): Theoretically, consider the case where symmetry is achieved in the anchored distributions (i.e., priors are equal to anchors), that is, both directions define the same P(x,z). In this case, take the equation H(x,z) = H(x) + H(z) - I(x;z). We know that without symmetry, neither H(x) or H(z) is constant with respect to the parameters. In the encoding case it's H(x), and in the decoding case it's H(z).
> However, when symmetry is achieved, then both distributions will share the same marginals. That is, P_enc(x) = P_dec(x) and P_enc(z) = P_dec(z). This means that we can replace H_enc(z) with constant H_dec(z), and analogously H_dec(x) with the constant H_enc(x) (remember, they are constant because of the anchors).
> In summary, this means that, effectively, H(x, z) = const + const - I(x; z). This gives another motivation why symmetry is important: it allows us to directly target mutual information.
>
> In practice, even when symmetry is not fully achieved, the symmetry regularizer R_theta actively encourages a solution with high mutual information,
> since H(x) and H(z) are very much constrained by P(x), P(z), correspondingly (see Eq. 11 for the explicit interpretable terms).
>
> Further, all things being equal, with MI fixed, one would prefer lower entropy models (i.e., no spurious information in z that is not directly relevant to x).

---

> > ### Public Comment · ~Jianlin_Su1 · 2019-10-22
> > **Is there any result on cifar10?**
> >
> > OK, I see. But your mutual information is a "global" mutual information of x and z. But deep infomax (https://arxiv.org/abs/1808.06670) show us that the "local" mutual information is actually more importance on complex images(e.g. cifar10). So I think you should introduce some local objects like local mutual information.

---

> > > ### Author Response · Authors · 2019-10-22
> > > **We followed the experimental setup in VampPrior paper which does not include color images.**
> > >
> > > Thanks for your suggestion. We will explore this in future work.

---

### Decision · Program_Chairs · 2019-12-19

**Decision:**

Reject

**Comment:**

The paper proposes an autoencoder framework for learning joint distributions over observations and latent states. The reviewers expressed concerns regarding the motivation for this work, the presentation with respect to prior work, and unconvincing experiments. In its current form the paper is not ready for acceptance to ICLR-2020.